# Chitosan functionalized Mn$_3$O$_4$ nanoparticles counteracts ulcerative colitis in mice through modulation of cellular redox state

Susmita Mondal[1], Monojit Das[2,3], Ria Ghosh[1], Manali Singh[4], Aniruddha Adhikari [1], Soumendra Darbar[5], Anjan Kumar Das[6], Siddhartha Sankar Bhattacharya[2], Debasish Pal[2], Debasish Bhattacharyya[7], Ahmed S. A. Ahmed[8], Asim Kumar Mallick[9], Munirah M. Al-Rooqi[10], Ziad Moussa[11], Saleh A. Ahmed [10,12✉] & Samir Kumar Pal [1,2✉]

Recent findings suggest a key role for reactive oxygen species (ROS) in the pathogenesis and progression of ulcerative colitis (UC). Several studies have also highlighted the efficacy of citrate functionalized Mn$_3$O$_4$ nanoparticles as redox medicine against a number of ROS-mediated disorders. Here we show that synthesized nanoparticles consisting of chitosan functionalized tri-manganese tetroxide (Mn$_3$O$_4$) can restore redox balance in a mouse model of UC induced by dextran sulfate sodium (DSS). Our in-vitro characterization of the developed nanoparticle confirms critical electronic transitions in the nanoparticle to be important for the redox buffering activity in the animal model. A careful administration of the developed nanoparticle not only reduces inflammatory markers in the animals, but also reduces the mortality rate from the induced disease. This study provides a proof of concept for the use of nanomaterial with synergistic anti-inflammatory and redox buffering capacity to prevent and treat ulcerative colitis.

[1] Department of Chemical, Biological Sciences, S. N. Bose National Centre for Basic Sciences, Block JD, Sector 3, Salt Lake, Kolkata 700106, India. [2] Department of Zoology, Uluberia College, University of Calcutta, Uluberia, Howrah 711315, India. [3] Department of Zoology, Vidyasagar University, Rangamati, Midnapore 721102, India. [4] Department of Biotechnology, Thapar Institute of Engineering and Technology, Bhadson Road, Patiala, Punjab 147004, India. [5] Research & Development Division, Dey's Medical Stores (Mfg.) Ltd, 62, Bondel Road, Ballygunge, Kolkata 700019, India. [6] Department of Pathology, Cooch Behar Government Medical College & Hospital, Vivekananda Rd, Khagrabari, Cooch Behar, West Bengal 736101, India. [7] Department of Gynecology & Obstetrics, Nil Ratan Sircar Medical College & Hospital, 138, AJC Bose Road, Sealdah, Raja Bazar, Kolkata 700014, India. [8] Faculty of Medicine, Assiut University, 71516 Assiut, Egypt. [9] Department of Pediatric Medicine, Nil Ratan Sirkar Medical College and Hospital, 38, Acharya Jagadish Chandra Bose Rd, Sealdah, Raja Bazar, Kolkata, West Bengal 700014, India. [10] Department of Chemistry, Faculty of Applied Science, Umm Al-Qura University, 21955 Makkah, Saudi Arabia. [11] Department of Chemistry, College of Science, United Arab Emirates University, P.O. Box 15551 Al Ain, United Arab Emirates. [12] Department of Chemistry, Faculty of Science, Assiut University, 71516 Assiut, Egypt. ✉email: saahmed@uqu.edu.sa; skpal@bose.res.in

Ulcerative colitis, a chronic idiopathic gastrointestinal disease characterized by a dysregulated immune response and persistent inflammation in the colonic and rectal mucosae[1], is the dominant subtype (Crohn's disease being the other) of inflammatory bowel disease (IBD) having a prevalence of ~0.25% of the population in North America, Europe, and Oceania[2]. The direct and indirect annual healthcare expenditures associated with ulcerative colitis are estimated to be as high as USD 12.8–29.8 billion in Europe and USD 8.1–14.9 billion in the United States[3]. With no proper cure, ulcerative colitis often poses an increased risk of numerous digestive and malignant pathologies (e.g., colorectal cancer) associated with uncontrolled inflammation leading to significant disability-adjusted life years, morbidity, and mortality[4,5]. The available conventional treatment options are 5-amino aminosalicylic acid (ASA), corticosteroids, immunosuppressants (e.g., azathioprine, 6-mercathioprine, and methotrexate), and molecular targeting of inflammatory cascades (e.g., anti-TNF-α drugs infliximab, adalimumab, and golimumab)[6–9]. For mild to moderate disease 5-ASA is effective but for severe cases, 5-ASA has been proved to be ineffective[8,10,11]. Administration of corticosteroids (intravenous or oral) is practiced for severe cases of UC[8]. The use of corticosteroids can provide primary remission for some cases but cannot completely remediate the UC. Previous reports on corticosteroid use suggested the involvement of many genetic factors regarding its clinical efficacy[12,13]. The other mentioned options are inadequate, expensive, have potentially life-threatening side effects, and can only induce clinical remission (not cure)[4,14–16]. Thus, it is of considerable interest to develop newer therapeutic modalities preferably with higher biocompatibility and lesser off-target adverse effects.

The etiology of ulcerative colitis is complex, multifaceted, and exposome dependent involving genetic predisposition, epithelial barrier defects, dysbiosis of gut microflora, dysregulated immune response, irregular lifestyle (smoking, high-sugar and high-fat diet, stress, etc.) and environmental factor[3,17]. In conjunction with the inappropriate activation of pro-inflammatory signaling cascades, recent findings insinuated the key involvement of reactive oxygen species (ROS) in the pathogenesis and progression of ulcerative colitis[18,19]. During the disease phase, infiltrated leukocytes, on activation, generates excess ROS (along with a wide spectrum of pro-inflammatory cytokines)and shifts the redox equilibrium of the intestinal mucosa toward oxidative distress, which in turn maintains active inflammation through induction of redox-sensitive signaling pathways and transcription factors[20]. Additionally, numerous pro-inflammatory molecules generate further oxidation products, leading to a self-sustaining and auto-amplifying vicious cycle that eventually disrupts the intestinal epithelial barrier[18]. Interestingly, both excessive and insufficient ROS contributes to the pathophysiology of ulcerative colitis[20]. For instance, a recent clinical study identified defective superoxide ($O_2\bullet^-$) generation as one of the major risk factors in intestinal inflammation[21]. Considering the causal relationship between ROS and inflammation in the disease pathogenesis, we hypothesized that a combination strategy involving the scavenging of ROS and the downregulation of pro-inflammatory mediators would be beneficial for the treatment of ulcerative colitis. However, trials with conventional non-selective small molecule antioxidants at high doses have generally failed in the prevention or treatment of ROS-related disease processes, likely owing to their inability to reach sufficient stoichiometric concentration inside cells or failure to maintain physiological ROS level(i.e., oxidative eustress) leading to disruption of vital redox-signaling pathways that controls processes like cellular metabolism, energetics, and survival[22]. One of the emerging solutions to this unique paradox about the pleiotropic role of ROS in disease progression is the introduction of redox-modulating nanomaterials with potential electron-donating as well as accepting capability to maintain precise balance between cellular oxidative eustress and distress (i.e., redox buffering; see ref. [23] for details). In recent years, some inorganic nanoparticles with anti-oxidant properties showed some promising results in the management of UC. Au and Zn nanoparticles achieved significant success by exhibiting anti-oxidant properties[24–26]. So, it can be hypothesized that advancing nanotechnology can provide a more successful way forward to the remission of UC via balancing the cellular redox state. In this regard, ligand-functionalized biocompatible tri-manganese tetroxide ($Mn_3O_4$) nanoparticles could be a practical choice. Several contemporary studies illustrated the efficacy of citrate functionalized $Mn_3O_4$ nanoparticles as redox medicine against a number of ROS-mediated disorders like neurodegenerative diseases[27,28], hepatic fibrosis[29], chronic kidney disease[30], and neonatal hyperbilirubinemia[31]. In addition, our recent study has shown how the ROS generation ability of folate-functionalized $Mn_3O_4$ nanoparticles could be used in the photodynamic therapy of adenocarcinoma[32]. Thus, this transition metal oxide nanomaterial is an ideal example of a redox buffer not only because of its ability to act as a catalytic antioxidant to protect mitochondria, the master regulator of cellular redox homeostasis[27] but also for its micro-environment dependent pro-oxidant activity[23]. Moreover, the nanomaterial is biocompatible and does not exert any detrimental effect on cellular components or biomolecules[23].

In this study, our primary aims were (i) to synthesize a biocompatible ligand (i.e., chitosan) functionalized $Mn_3O_4$ nanoparticle (Ch-$Mn_3O_4$ NP) that can efficiently function as cellular redox buffer; (ii) to evaluate its safety and efficacy as a potential redox medicine against ulcerative colitis in a well-established animal model i.e., dextran sulfate sodium (DSS) intoxicated BALB/c mice; and iii) to understand the underlying mechanism of therapeutic action. The choice of surface functionalizing ligand i.e., chitosan (a polysaccharide composed of randomly arranged β-linked D-glucosamine and N-acetyl-D-glucosamine[33]) was primarily dictated by (a) its stability in the pH range of gastrointestinal tract making it suitable for oral administration[34]; (b) biocompatibility with $LD_{50}$ value close to sugar (16 g kg$^{-1}$)[35]; and (c) specific and easy absorption in the colon due to the interaction between its positively charged primary amines and the negatively charged sialic acid substructures present in colonic mucus[36,37]. Here, we have successfully synthesized Ch-$Mn_3O_4$ NPs, thoroughly characterized their physicochemical properties using electron microscopic and UV–vis spectroscopic techniques, and evaluated in vitro redox buffering capacity (i.e., pro-oxidant and antioxidant activities). In animal experiments, clinical and histopathological signs of ulcerative colitis were markedly reduced after treatment with Ch-$Mn_3O_4$ NPs. Further molecular studies illustrated a significant reduction in intestinal pro-inflammatory cytokines, decreased macrophage activity, and preservation of cellular redox homeostasis through mitochondrial protection. A 28-day repeated dose sub-chronic toxicity study demonstrated the biocompatibility and safety of this treatment strategy. In summary, our preclinical animal study provides a proof of concept for the use of a nanomaterial (i.e., Ch-$Mn_3O_4$ NPs) with synergistic anti-inflammatory and redox buffering capacity as redox medicine to prevent and treat ulcerative colitis.

## Results

### Development of an orally administrable nanomaterial (i.e., Ch-$Mn_3O_4$ NPs) to target colon. The surface functionalizing ligands, size, and surface charge are three important factors that determine the specific distribution pattern of a nanomaterial inside

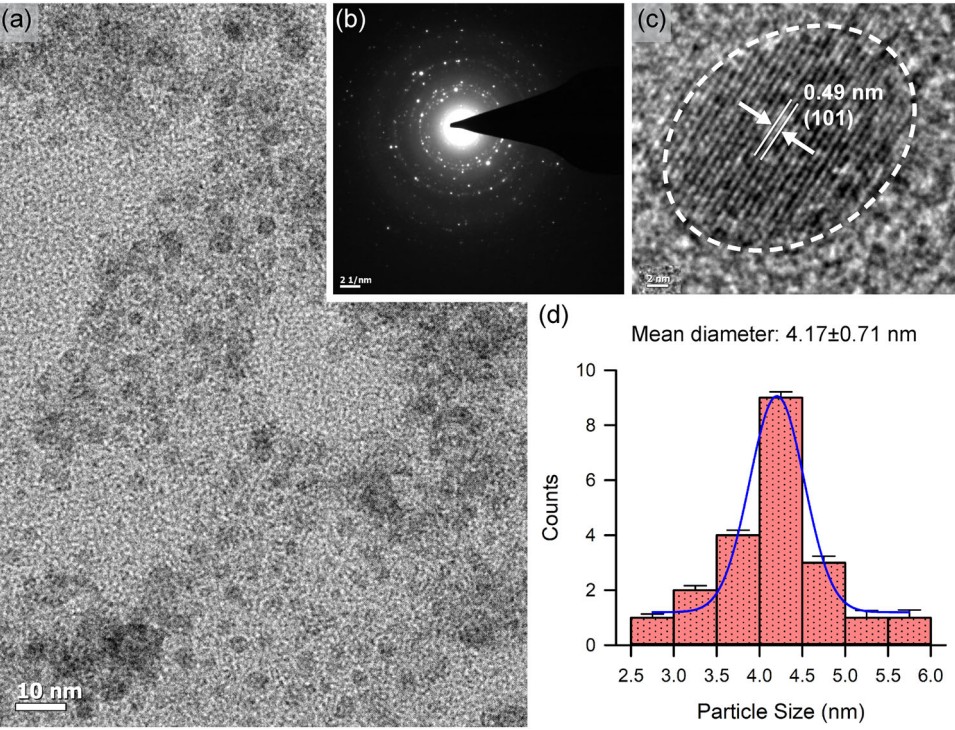

**Fig. 1 Physico-chemical characterization of Ch-Mn$_3$O$_4$ NPs. a** Transmission electron microscopy (TEM) image of Ch-Mn$_3$O$_4$ NPs. **b** SAED pattern of the same. **c** High-resolution TEM (HRTEM) image of a single nanoparticle. **d** Size distribution of Ch-Mn$_3$O$_4$ NPs as measured from TEM.

living organisms. Here, we have functionalized Mn$_3$O$_4$ NPs with a biocompatible ligand chitosan considering the prior knowledge about its well-known ability to specifically target and be absorbed in the colonic tissues. Covalent binding between ligand chitosan and the Mn$_3$O$_4$ NPs was confirmed from the observed perturbation in Fourier transform infrared spectra [Supplementary note, Supplementary Fig. 1] corresponding to C–N stretching at 1260 cm$^{-1}$of chitosan and surface OH groups at 1377 cm$^{-1}$ of the NPs. UV–vis spectroscopic studies [Supplementary Fig. 2] revealed signatures of ligand-to-metal charge transfer from chitosan to Mn$^{2+}$/Mn$^{3+}$of the NPs, the degeneracy of ground state d4 (Mn$^{3+}$) electrons (e.g., Jahn–Teller effect) and multiple photoluminescence at room temperature, the unique features otherwise absent in pristine non-functionalized Mn$_3$O$_4$ NPs (see Supplementary notes for specifics, Supplementary Fig. 2). The transmission electron micrographs (TEM) show the spherical shape and homogeneous distribution of Ch-Mn$_3$O$_4$ NPs [Fig. 1a]. The selected area electron diffraction (SAED) pattern shows the diffraction rings corresponding to (101) planes [Fig. 1b], confirming the spinel hausmannite structure of Ch-Mn$_3$O$_4$ NPs[38]. The inter-fringe distance from high-resolution TEM (HRTEM) was determined to be 4.9 Å [Fig. 1c], corresponding to the (101) plane of Mn$_3$O$_4$ crystal lattice[39]. Figure 1d suggests monomodal size distribution of spherical Ch-Mn$_3$O$_4$ NPs with a median diameter of 4.17 ± 0.71 nm. It has to be noted that, specific care was taken during the optimization of the synthesis method to confine the size of the NPs within 8.0 nm (the majority of the particles were in the 2.5–6.0 nm size range) as a diameter < 8.0 nm facilitates the elimination of NPs through renal excretory system, one of the major determinants of biocompatibility[40]. To check the effect of pH on the structure of the NPs, we have performed the TEM study at different pH values (Supplementary Fig. 3c–f). It is evident from the TEM results that the variation in pH does not significantly affect the overall shape or size of the NPs. The aqueous stability of Ch-Mn$_3$O$_4$ NPs, as determined by sequential monitoring of absorbance at 280 nm,

was found to be excellent (pH 2.5–6.5) (Supplementary Fig. 3a). But at higher pH (pH ~7.5) the NPs were precipitating out from the aqueous solution. This is due to the tendency of the chitosan molecule to be aggregated at higher pH. Moreover, the pH-dependent zeta potential study showed a similar trend (Supplementary Fig. 3b). At pH more than 7, Ch-Mn$_3$O$_4$ NPs were found to be aggregated and precipitate out from the solution. Due to this characteristic, Ch-Mn$_3$O$_4$ NPs become efficient ligands for colon-targeted drugs as at colonic pH (~7.2) it will be separated out from the solution and will adhere to the colonic mucosal layer[41].

**Ch-Mn$_3$O$_4$ NPs function as a redox buffer.** In order to maintain the cellular oxidative eustress condition, rapid sensing and adjustment of the redox environment are essential for redox buffering. According to our previous studies, Ch-Mn$_3$O$_4$ NPs have the ability to function as a redox buffering agent by sensing the cellular redox state and swiftly shifting the redox equilibrium to the eustress condition[23]. This behavior of Ch-Mn$_3$O$_4$ NPs as a redox modulator is analogous to that of pH buffers, where the pH buffering system can adjust the pH of a solution by sensing the H$^+$ concentration of the medium. In this study, by utilizing the 2′,7′-dichloro-dihydro-fluorescein (DCFH) assay we evaluated the pro-oxidant activity of Ch-Mn$_3$O$_4$ NPs (Supplementary Fig. 4a). Next, we investigated the radical scavenging activity of the NPs using ABTS [2,2′-azino-bis(3-ethylbenzothiazoline-6-sulfonic acid)] assay [Supplementary Fig. 4b]. The Ch-Mn$_3$O$_4$ NPs can effectively scavenge the ABTS•$^+$ radical cation (0.5 μg ml$^{-1}$ NPs showed anti-oxidant potential equivalent to 0.43 mM Trolox solution, water-soluble vitamin E analog). After getting insight into the ROS generation and radical scavenging activity of the NPs, we evaluated the in vitro redox buffering capacity of the synthesized NPs using H$_2$O$_2$ as an oxidant molecule. With the increase in the concentration of H$_2$O$_2$, the rate of DCFH oxidation increased within the medium [Supplementary Fig. 4c]. However, on the introduction of Ch-Mn$_3$O$_4$ NPs in the active medium, a significant alteration in the DCFH

intensity signifies the quenching of the ROS generated by the $H_2O_2$, acting as an oxidant in the medium. It is worthwhile to mention that Ch-$Mn_3O_4$ NPs always maintained a base level of ROS concentration within the reaction medium, rather than eliminating all the ROS molecules. Prior findings have shown that a strong relationship exists between the intracellular $H_2O_2$ concentration and cellular oxidative state, and regular cellular functions are maintained up to 0.1 μM $H_2O_2$ concentration, beyond that triggers detrimental pathways that lead to cell death. Here, we have determined the concentration of Ch-$Mn_3O_4$ NPs equivalent to the intracellular concentration of $H_2O_2$ in terms of ROS generation [Supplementary Fig. 4d]. Supplementary Fig. 4e shows the cell response pattern associated with intracellular Ch-$Mn_3O_4$ NP concentration. In summary, the in vitro studies clearly describe the ability of Ch-$Mn_3O_4$ NPs to maintain oxidative eustress, i.e., persistent maintenance of the cellular ROS level for proper physiological functioning (oxidative eustress) while evading the excess ROS from the medium (oxidative distress).

**Ch-$Mn_3O_4$ NPs improved clinical symptoms in animal models of acute ulcerative colitis.** Promising redox buffering capability and favorable physicochemical properties of Ch-$Mn_3O_4$ NPs propelled us to move forward and initiate preclinical animal studies. Here, we evaluated the treatment outcomes of Ch-$Mn_3O_4$ NPs in DSS-induced BALB/c mice, a well-established animal model for testing therapeutic interventions against ulcerative colitis[42]. Severely elevated disease activity index (DAI), a combined score comprising of three prominent clinical parameters—weight loss, stool consistency, and fecal blood content[43], in DSS (5% in drinking water) intoxicated mice (DAI: $9.60 \pm 0.96$ compared to $0.0 \pm 0.0$ of control littermates, $p < 0.0001$, one-way analysis of variance (ANOVA), $F_{(3, 24.24)} = 310.0$) at day 12 indicates successful induction of ulcerative colitis [Fig. 2a]. Drastic reduction in body weight (BW change: $-12.70 \pm 1.91\%$ in the DSS-intoxicated group compared to $10.81 \pm 2.74\%$ in control, $p < 0.0001$, one-way ANOVA, $F_{(3, 24.13)} = 205.2$) further supports acute colitis induction [Fig. 2b]. For the therapeutic settings, treatments were administered daily starting on day 13, and final outcomes were assessed on day 28. Oral administration of Ch-$Mn_3O_4$ NPs ($0.25 \, \mathrm{mg \, kg^{-1}}$ BW, single dose daily) for 15 consecutive days significantly reduced the DAI ($0.80 \pm 0.42$ compared to $7.40 \pm 1.07$ of ulcerative colitis model cohort, $p < 0.0001$, one-way ANOVA, $F_{(3, 17.71)} = 290.8$) [Fig. 2a] and prevented body weight loss (BW change: $-2.31 \pm 2.69\%$ vs. $-15.69 \pm 2.71\%$ of ulcerative colitis model cohort, $p < 0.0001$, one-way ANOVA, $F_{(3, 34.93)} = 332.1$) [Fig. 2b]. The body weight changes were similar in the control and Ch-$Mn_3O_4$ NPs control cohort (BW change: $18.51 \pm 2.91\%$ vs. $18.50 \pm 3.30\%$ of control cohort, $p = 0.9941$, one-way ANOVA, $F_{(3, 34.93)} = 332.1$) [Fig. 2b]. As bloody diarrhea is one of the primary and well-reported symptoms of acute colitis[43,44], we monitored the fecal blood content throughout the experimental period. On day 12, the DSS-intoxicated mice showed a large amount of blood in their feces [Fig. 2c, d]. While the fecal blood content remained high in the ulcerative colitis model cohort for the rest of the treatment period (days 13–28), in Ch-$Mn_3O_4$ NPs treated cohort it started to reduce from day 20 and disappeared on day 28 [Fig. 2c, d]. Bloody diarrhea was accompanied by reduced colon length, another typical characteristic of acute colitis, in the ulcerative colitis model (colon length: $4.80 \pm 0.55$ cm vs. $8.42 \pm 0.39$ cm of control cohort, $p < 0.0001$, one-way ANOVA, $F_{(3, 32.88)} = 105.1$) [Supplementary Fig. 5]. Treatment with Ch-$Mn_3O_4$ NPs significantly improved the colon length comparable to control mice (colon length: $8.00 \pm 0.62$ cm vs. $8.42 \pm 0.39$ cm of control cohort, $p = 0.09$, one-way ANOVA, $F_{(3, 32.88)} = 105.1$)

[Supplementary Fig. 5]. Also, treatment of NPs reduces the significant mortality rate from the induced disease [Supplementary Fig. 6].

**Ch-$Mn_3O_4$ NPs bolstered histological healing and the formation of regenerative glands.** Along with the resolution of clinical symptoms (i.e., cessation of rectal bleeding and improvement in bowel habits), the primary endpoint of all treatment strategies for ulcerative colitis comprises histopathological healing (i.e., amelioration of mucosal damages and inflammation) as they improve long-term remission, decrease the risk of colectomy, and limit corticosteroid use[45,46]. In this study, hematoxylin and eosin (H&E)-stained colon tissue sections isolated from both control and Ch-$Mn_3O_4$ NPs control cohorts show normal histological features of the colon like healthy mucosal and submucosal layer, typical muscularis propria and serosa [Fig. 3a, d]. In contrast, the colon tissue section from the ulcerative colitis model cohort shows prominent pathological features of ulcerative colitis like well-demarcated superficial mucosal ulceration and massive infiltration of inflammatory cells within lamina propria. In addition, the presence of severe cryptitis and crypt abscesses, a moderate increase in the number of neutrophils in lamina propria along with the presence of small areas of granulation tissues and mucosal fibrosis in lamina propria indicate severe architectural abnormalities in the colon (Fig. 3b). Treatment with Ch-$Mn_3O_4$ NPs for 15 days lead to remarkable improvements in the histopathological features of the damaged colon. The extent of inflammatory cell infiltration reduced significantly along with normal-looking crypt cells, the usual number of neutrophils in lamina propria and epithelium, and regenerating epithelial layer with a negligible level of adjacent inflammation. The H&E-stained colon section from this group [Fig. 3c] clearly shows the healed ulcerated area covered by regenerating epithelium along with several regenerative glands, indicating the initiation of regenerative pathways. In brief, the colon from the Ch-$Mn_3O_4$ NPs treated group shows an almost healthy mucosal layer, muscularis propria, submucosal layer, and serosa with mild abnormalities. To get an objective, semiquantitative, reproducible depiction of the histopathological healing described in the above section, we have evaluated the histology score of H&E stained colon tissue sections according to the Geboes Score, the most commonly used histological score in ulcerative colitis (Supplementary Fig. 7)[46,47]. The Geboes Score for the ulcerative colitis model cohort was significantly high indicating the severity of colonic damage i.e., architectural abnormalities, chronic inflammatory cells infiltration, presence of neutrophils in lamina propria and epithelium, destruction of the crypts and presence of ulcer and granulation tissue (Geboes score: $14.00 \pm 0.81$ vs. $0.10 \pm 0.31$ of control cohort, $p < 0.0001$, one-way ANOVA, $F_{(3, 22.10)} = 844.7$) [Supplementary Fig. 7]. Treatment with Ch-$Mn_3O_4$NPs decreased the score within the normal range (Geboes score: $3.00 \pm 1.00$ vs. $14 \pm 0.81$ of ulcerative colitis model cohort, $p < 0.0001$, one-way ANOVA, $F_{(3, 22.10)} = 844.7$) which is the confirmation of the histological healing of the colonic lesions, reduced inflammation and restoration of architectural integrity [Supplementary Fig. 7]. The Geboes Score for control and Ch-$Mn_3O_4$ NPs control cohorts were similar ($0.20 + 0.42$ vs. $0.10 + 0.31$ of control cohort, $p = 0.5566$, one-way ANOVA, $F_{(3, 22.10)} = 844.7$) [Supplementary Fig. 7]. So, treatment with Ch-$Mn_3O_4$ NPs did not cause any kind of architectural abnormality in the colon of Ch-$Mn_3O_4$ NPs control cohorts (Fig. 3d). For a better understanding of the healing process, histopathological images of all four groups in lower magnification are provided in Supplementary Fig. 8.

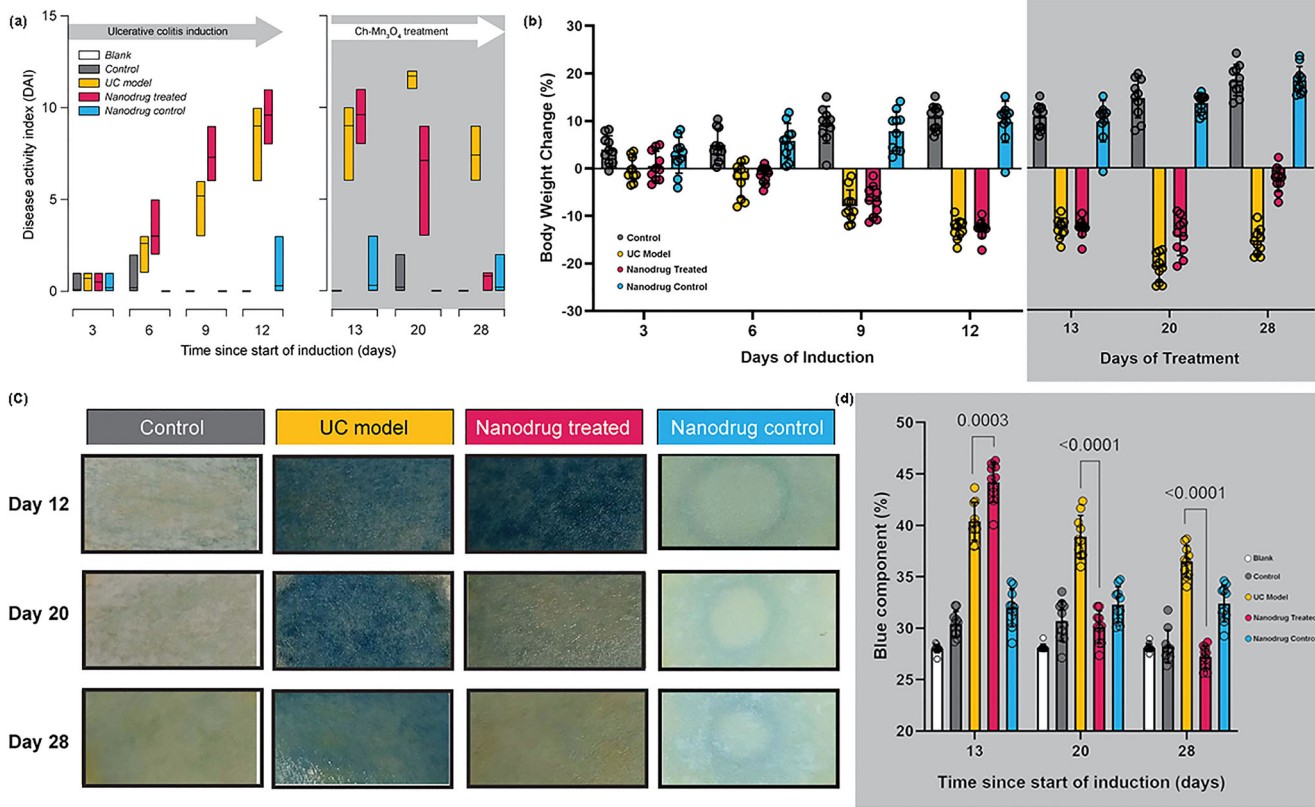

**Fig. 2 Evaluation of clinical symptoms of ulcerative colitis. a** DAI score throughout the ulcerative colitis induction and the treatment phase. Treatment with Ch-Mn$_3$O$_4$ NPs successfully prevented as well as reduced the DAI score to a normal level. **b** Body weight change across all the groups during the induction and Ch-Mn$_3$O$_4$ NPs treatment phase. **c** Picture of OBT test strips. **d** Fecal blood indices (blue component analysis of OBT test strips). The amount of blood in feces significantly reduced after treatment with Ch-Mn$_3$O$_4$ NPs. In bar plots, data were expressed as mean ± SD ($n = 10$). Individual data points are represented as colored circles ($n = 10$). One-way analysis of variance (ANOVA) followed by correction of false discovery rate (post hoc FDR: two-stage step-up method of Benjamini, Krieger, and Yekutieli) for multiple comparisons was performed for comparison between multiple groups. $p < 0.05$ is considered significant.

**Ch-Mn$_3$O$_4$ NPs reduce markers of inflammation: pro-inflammatory cytokines and macrophage infiltration.** Activation of pro-inflammatory cytokines particularly IL-1β and IL-12 play a pivotal role in the induction, pathogenesis, and relapse of ulcerative colitis[48–50]. As illustrated in Fig. 4, serum levels of both cytokines were severely elevated representing the induction of intense inflammation. In agreement with the histopathological observations, administration of Ch-Mn$_3$O$_4$ NPs drastically reduced the levels of both IL-1β (IL-1β: 447.7 ± 112.3 pg ml$^{-1}$; $p < 0.0001$, $F(3, 15.8) = 786.6$, one-way ANOVA, compared to diseased model IL-1β: 2882 ± 252.4 pg ml$^{-1}$) Fig. 4a and IL-12 (IL-12: 34.17 ± 3.49 pg ml$^{-1}$; $p < 0.0001$, $F(3, 33.10) = 417.2$, one-way ANOVA, compared to diseased model IL-12: 72.45 ± 4.16 pg ml$^{-1}$) Fig. 4b. Neither of the other two experimental cohorts show signs of inflammation.

Macrophages are one of the most abundant leukocytes found in luminal gastrointestinal mucosa and play a vital role in maintaining intestinal mucosal homeostasis[51]. Infiltration of macrophages in lamina propria and subsequent induction of pro-inflammatory cascades is a major event in the progression of both ulcerative colitis and Crohn's disease[52,53]. Prior studies have also proposed potential benefits for the therapeutic targeting of macrophages in IBD[54]. Here, we stained the colon tissue sections with an anti-CD68 antibody, a popular marker for M1 macrophages, to visualize the level of macrophage infiltration. In the control cohort, accumulation of CD-68$^+$ macrophages was seen beneath the colonic mucosa forming a dense sub-epithelial band and only a small fraction of CD-68$^+$ macrophages were

present in the lamina propria [Fig. 4c]. In the colitis model cohort, a hefty number of CD-68$^+$ macrophages were detected in both the sub-epithelial and lamina propria regions [Fig. 4c]. Such infiltration of macrophages reflected in the observed large CD-68$^+$ positive region [Fig. 4c]. The Ch-Mn$_3$O$_4$ NPs treated cohort showed minimal CD-68$^+$ macrophages in the lamina propria and the lesser dense subepithelial band even compared to the control ones. The CD-68$^+$ positive region was also smaller in this group. Ch-Mn$_3$O$_4$ NPs control cohort showed similar staining to the control group.

**Ch-Mn$_3$O$_4$ NPs inhibit inflammasome-mediated Caspase-1 activation and down-regulate NF-κβ expression in colonic tissue.** To get further insight into the upstream regulators of the anti-inflammatory phenomena displayed by Ch-Mn$_3$O$_4$ NPs, we investigated the expression profiles of two key proteins namely Caspase-1, responsible for cell death[55,56], and NF-κβ, a major transcription factor regulating the pro-inflammatory cytokines[57–59]. The strong green fluorescence from the cytosolic regions of the colon tissues stained with anti-Caspase-1 antibody from ulcerative colitis model mice clearly reveals high expression of Caspase-1 (Mean Fl. intensity: 46.36 ± 2.09; $p < 0.0001$, $F(3, 33.61) = 557.1$, one-way ANOVA, compared to control Mean Fl. intensity: 32.86 ± 1.98) [Fig. 4c] compared to the control mice. In contrast, Caspase-1 expression was much lesser in Ch-Mn$_3$O$_4$ NPs treated group than the diseased ones as illustrated by the lower intensity of the green fluorescence (Mean Fl. intensity: 24.77 ± 1.64; $p < 0.0001$, $F(3, 33.61) = 557.1$, one-way ANOVA,

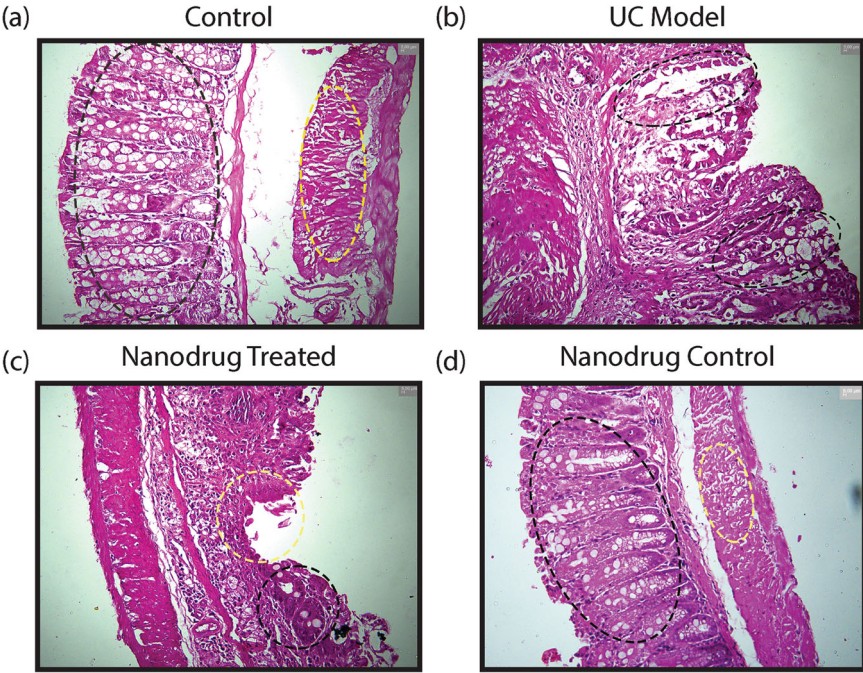

(a) Control (b) UC Model

(c) Nanodrug Treated (d) Nanodrug Control

**Fig. 3 Efficacy of Ch-Mn$_3$O$_4$ NPs in a reversal of structural abnormality in the animal model of ulcerative colitis. a** Micrographs of hematoxylin and eosin stained colon section of control mice. Black oval: mucosal layer; yellow oval: muscularis propria. **b** Micrographs of hematoxylin and eosin stained colon section of DSS intoxicated diseased group. Black oval: infiltration of inflammatory cells. **c** Micrographs of hematoxylin and eosin stained colon section of Ch-Mn$_3$O$_4$ NPs treated group. The colon section of Ch-Mn$_3$O$_4$ NPs treated mice shows the healed ulcerated area covered by regenerating epithelium and other important glands (black circle: regenerative glands; yellow circle: healing region). **d** Micrographs of hematoxylin and eosin stained colon section of Ch-Mn$_3$O$_4$ NPs control group. Black oval: mucosal layer; yellow oval: muscularis propria.

compared to diseased model Mean Fl. intensity: $46.36 \pm 2.09$). Similarly, we monitored the expression profile of NF-κβ in colon tissues using fluorescent-tagged anti-NF-κβ p65 subunit antibody. In the ulcerative colitis model group, the presence of NF-κβ was abundant in both cytosol and nuclei of the colonic epithelial cells compared to the control one [Fig. 4c] indicating upregulation of this transcription factor (Mean Fl. intensity: $42.54 \pm 2.17$; $p < 0.0001$, $F_{(3, 34.48)} = 322.3$, one-way ANOVA, compared to control Mean Fl. intensity: $22.98 \pm 1.88$). Whereas, the Ch-Mn$_3$O$_4$ NP treated group showed much lower fluorescence intensity (Mean Fl. intensity: $28.65 \pm 2.19$; $p < 0.0001$, $F_{(3, 34.48)} = 322.3$, one-way ANOVA, compared to diseased model Mean Fl. intensity: $42.54 \pm 2.17$) indicative of down-regulated NF-κβ expression.

**Ch-Mn$_3$O$_4$ NPs mitigate redox-mediated inflammation through the protection of mitochondria from oxidative damages.** Recent evidence suggests that mitochondria lie at the heart of immunity and mitochondrial alterations play a central role in the pathogenesis of a plethora of inflammation-mediated acute and chronic diseases including ulcerative colitis[60,61]. Oxidative damage to intestinal mitochondria may lead to the initiation of the NF-κβ pathway, formation of NRLP3 inflammasome, and caspase-1 activation[60,62]. Therefore, we evaluated the mitochondrial 'health' in different experimental cohorts. The results clearly suggest that induction of ulcerative colitis triggered detrimental alterations to important mitochondrial health indicators resulting in degenerative redox signaling. Induction of ulcerative colitis led to increased mitochondrial membrane permeabilization i.e., mitochondrial swelling or mPTP formation. The time-dependent sharp decrease in the 540 nm absorbance [Fig. 5a] of mitochondria isolated from ulcerative colitis model cohort upon Ca$^{2+}$ sensitization is a clear manifestation of this phenomenon. This was accompanied by deregulated mitochondrial membrane potential ($\Delta\Psi_m$: $68.81 \pm 4.57\%$; $P < 0.0001$, $F_{(3, 27.56)} = 178$,

one-way ANOVA, compared to control $\Delta\Psi_m$: $100 \pm 3.02\%$) [Fig. 5b], decreased ATP level (ATP content: $16.00 \pm 2.40$ nmol mg$^{-1}$ protein; $P < 0.0001$, $F_{(3, 35.55)} = 127.9$, one-way ANOVA, compared to control ATP content: $33.17 \pm 2.30$ nmol mg$^{-1}$ protein) [Fig. 5c], and increased mitochondrial ROS formation (mt ROS: $139.3 \pm 6.25\%$; $p < 0.0001$, $F_{(3, 29.22)} = 140.5$, one-way ANOVA, compared to control mt ROS: $100 \pm 2.33\%$) [Fig. 5d]. These alterations were abrogated by Ch-Mn$_3$O$_4$ NP treatment; regulated mitochondrial membrane potential ($\Delta\Psi_m$: $92.72 \pm 3.89\%$; $p < 0.0001$, $F_{(3, 27.56)} = 178$, one-way ANOVA, compared to ulcerative colitis model cohort $\Delta\Psi_m$: $68.81 \pm 4.57\%$), increased ATP level (ATP content: $28.91 \pm 2.18$ nmol mg$^{-1}$ protein; $p < 0.0001$, $F_{(3, 35.55)} = 127.9$, one-way ANOVA, compared to ulcerative colitis model cohort ATP content: $16.00 \pm 2.40$ nmol mg$^{-1}$ protein), decreased mitochondrial ROS formation (mt ROS: $105.4 \pm 5.50\%$; $p < 0.0001$, $F_{(3, 29.22)} = 140.5$, one-way ANOVA, compared to ulcerative colitis model cohort mt ROS: $139.3 \pm 6.25\%$) [Fig. 5a–d].

**Biocompatibility of Ch-Mn$_3$O$_4$ NPs.** We evaluated the biocompatibility of the prepared Ch-Mn$_3$O$_4$ NPs in in-cellulo model. In the first step, we used the human embryonic kidney (HEK 293) cell line, a normal epithelial cell line, for this purpose. First, we treated the HEK 293 cell line with three different concentrations ($0.05$ µg ml$^{-1}$, $0.5$ µg ml$^{-1}$, $5$ µg ml$^{-1}$) of NPs and incubated them for 6 h. Then using the MTT assay, we determined the cell viability. Figure 6a exhibits the cell viability for three different concentrations of NPs. It is evident from Fig. 6a that the cells that received $0.05$ µg ml$^{-1}$ and $0.5$ µg ml$^{-1}$ of NPs show similar cell viability as the control group. But the cells treated with $5$ µg ml$^{-1}$ of NPs showed significant cell viability (~70%) compared to control cells.

To understand this phenomenon of lower cell viability at higher concentration NPs, we have established a relation between

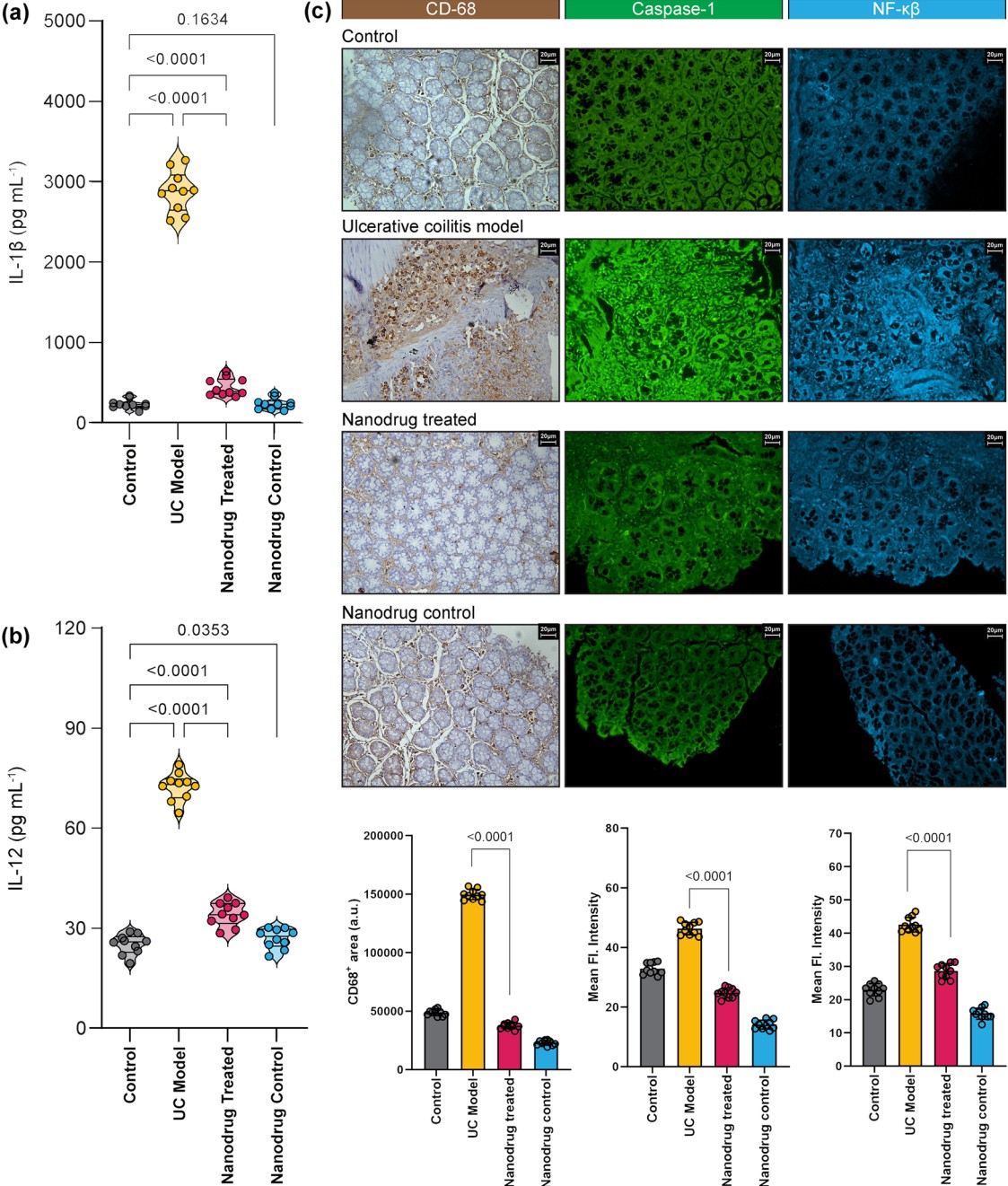

**Fig. 4 Evaluation of pro-inflammatory cytokines and Immunohistopathological changes of colon tissues collected from the control, DSS, DSS + Ch-Mn$_3$O$_4$ NPs, and Ch-Mn$_3$O$_4$ NPs control groups. a** Amount of IL-1β in blood samples collected from each study group. **b** Amount of IL-12 in blood samples collected from each study group. **c** Colonic tissue section stained with anti-CD-68 antibody. Treatment with Ch-Mn$_3$O$_4$ NPs shows mild no of positive cells indicating prevention of the infiltration of M1 macrophage leading to reduced inflammation. The section was also stained with an anti-caspase-1 antibody. Treatment with Ch-Mn$_3$O$_4$ NPs successfully prevents caspase-1 activation which indicates cessation of inflammasome assembly. Further section stained with anti-NF-κβ p65 antibody. Micrographs show a lower presence of the protein indicating the downregulation of free NF-κβ in mice treated with NPs. CD-68 positive area and mean fluorescence intensity in the case of anti-caspase-1 and anti-NF-κβ stained area were also mentioned. Violins depict kernel density estimation of the underlying data distribution with the width of each violin scaled by the number of observations at that *Y*-value. Three lines (from the bottom to the top) in each violin plot show the location of the lower quartile (25th), the median, and the upper quartile (75th), respectively. The shaded area indicates the probability distribution of the variable. Individual data points are represented as colored circles (*n* = 10). One-way analysis of variance (ANOVA) followed by correction of false discovery rate (post hoc FDR: two-stage step-up method of Benjamini, Krieger, and Yekutieli) for multiple comparisons was performed for comparison between multiple groups. The numbers inside the plots indicate numerical *p* values. *p* < 0.05 is considered significant.

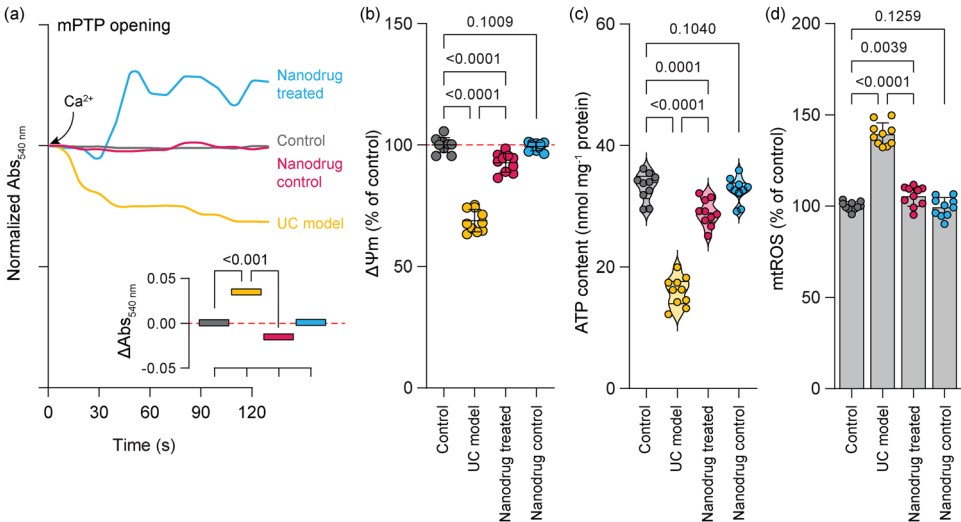

**Fig. 5 Effect of Ch-Mn₃O₄ NPs on mitochondrial parameters of the colon. a** Effect on mitochondria permeability transition, measured as a decrease in absorbance at 540 nm. **b** Change in mitochondrial membrane potential ($\Delta\Psi$m). **c** ATP content. **d** Mitochondrial ROS (mtROS) as measured using DCFH assay. Ch-Mn₃O₄ NPs efficiently protect the mitochondria from oxidative damage. In bar plots, data were expressed as mean ± SD ($n = 10$). Violins depict kernel density estimation of the underlying data distribution with the width of each violin scaled by the number of observations at that $Y$-value. Three lines (from the bottom to the top) in each violin plot show the location of the lower quartile (25th), the median, and the upper quartile (75th), respectively. The shaded area indicates the probability distribution of the variable. Individual data points are represented as colored circles ($n = 10$). One-way analysis of variance (ANOVA) followed by correction of false discovery rate (post hoc FDR: two-stage step-up method of Benjamini, Krieger, and Yekutieli) for multiple comparisons was performed for comparison between multiple groups. The numbers inside the plots indicate numerical $p$ values. $p < 0.05$ is considered significant.

the amount of ROS generated by a certain concentration of NPs and ROS produced by a certain concentration of $H_2O_2$. Supplementary Fig. 4d depicts that the concentration of NPs equivalent to the concentration of $H_2O_2$ in terms of ROS generation. The graph shows a linear relation. Previous studies have shown that a strong relationship exists between the intracellular $[H_2O_2]$ and the cell's oxidative state[22]. It is reported that the cells perform their normal function up to 0.1 μM of intracellular $H_2O_2$ concentration (oxidative eustress)[22]. An increase in intracellular $H_2O_2$ concentration beyond 0.1 μM triggers the detrimental pathways that lead to cell death (oxidative distress)[22] This intracellular concentration of $H_2O_2$ varies from cell type to cell type. From our calculation, we have created a similar cell response pattern with increasing intracellular NPs concentration (Supplementary Fig. 4e). This picture gives a comprehensible idea regarding the outcome of our cell viability assay. As the cells treated with 0.05 μg ml⁻¹ and 0.5 μg ml⁻¹ of NPs maintain oxidative eustress, no significant cell death was observed. However, the cells that received 5 μg ml⁻¹ of NPs showed significant cell death because at that NP concentration cells failed to retain their oxidative eustress condition and excess ROS generation damaged the cell components. So, from the above discussion, it is evident that these NPs can effectively buffer the redox state of the cells and help them to perform their normal physiological function. But beyond a certain level, it also creates oxidative distress due to its ROS generation ability at room temperature.

A hemolysis study (Fig. 6b) has also been performed using 0.5 μg ml⁻¹ of NPs. RBCs were incubated with NPs, water, and phosphate buffer saline (PBS) for 3 h. The study set with NPs no visible hemolysis like the PBS one. Further, the spectroscopic assessment of the supernatant also confirm that NPs did not exert any toxic effect on RBCs.

Further, we carried out the bio-distribution of the NPs in different organs using ICPOES (Fig. 6c). After 7 days of treatment, the manganese (Mn) content in a colon tissue sample

of Ch-Mn₃O₄ NPs treated mice (Mn content: 1.78 ± 0.15 mg kg⁻¹; $P < 0.0001$, $t = 14.05$, d$f = 17.22$; Unpaired $t$-test, compared to control Mn content: 0.92 ± 0.12 mg kg⁻¹) is significantly higher in comparison to the untreated control cohort. But there was no significant accumulation of NPs in any organs other than the colon. So, the results of the bio-distribution study suggest the high affinity of Ch-Mn₃O₄ NPs toward the colon which is the primary target organ of this study.

A sub-chronic toxicity study to evaluate the bio-compatibility and toxicity of Ch-Mn₃O₄ NPs in the preclinical mice model was performed (Fig. 6d). We have chosen three doses of NPs, 0.25 mg kg⁻¹ Body weight (BW), 0.5 mg kg⁻¹ BW and 1 mg kg⁻¹ BW. Three groups of mice were orally treated with a single dose of NPs on a daily basis. A fourth group of mice were treated with saline water (oral administration) which serves as a control study group. After 28 days, we collected organs from all the study groups and performed histological analysis to check any structural abnormalities in the organs of NPs treated mice. Figure 6d shows the histological structure of the colon, liver, spleen, and kidney of all four groups. No structural abnormality is observed in the histological examination of any organs of the NPs treated mice. Further, we have also evaluated the histopathological architecture of the heart and lungs of all the study groups. Figure 6d shows the normal histological structure of cardiac myocytes with well-placed nuclei all across the NPs treated mice. The lung sections also showed normal lung tissue with a fair no of alveoli. All other organs of NPs treated mice successfully retained their normal architecture as compared to the control study group. So, the histopathological analysis confirms that the use of NPs in mice did not cause any kind of toxicity at the tissue level also. Further, we performed an analysis of hematological parameters across all the groups to confirm the histopathological result at the biochemical level. Any kind of biochemical changes will reflect the effect on the hematological parameters. The results show all the parameters are in the normal range in all NPs treated groups (Supplementary Table S1). All the mice treated with different

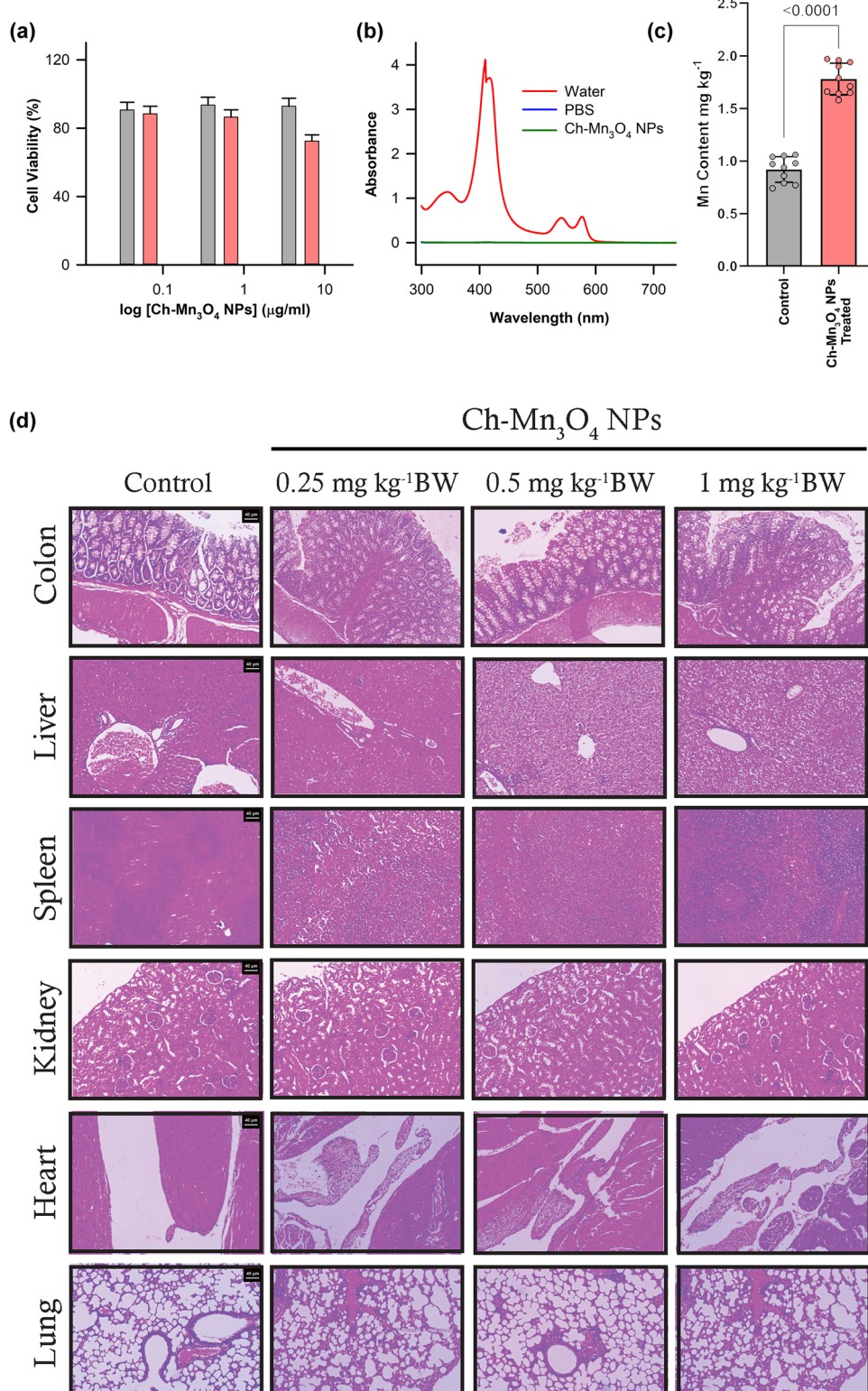

**Fig. 6 Bio-compatibility of Ch-Mn₃O₄ NPs. a** Cell-viability assay using HEK 293 cell line. **b** Hemolysis assay. **c** Mn-content of control mice and Ch-Mn₃O₄ treated mice. In bar plots, data were expressed as mean ± SD (n = 10). Individual data points are represented as colored circles (n = 10). An unpaired, parametric t-test with Welch's correction was performed. p < 0.05 is considered significant. **d** Sub-chronic toxicity of Ch-Mn₃O₄ NPs. Micrographs of hematoxylin and eosin stained sections of colon, liver, spleen, kidney, Lung and Heart after 28 days of treatment with Ch-Mn₃O₄ NPs at three dose levels i.e., 0.25 mg kg⁻¹ BW, 0.5 mg kg⁻¹ BW, and 1 mg kg⁻¹ BW. Colon sections show large guts comprising of healthy mucosa layer, muscularis propria, and submucosa layer across all groups. Control and all the NPs treated groups liver section shows normal looking central vein, portal tracts, hepatocytes arranged in cords with normal sinusoidal space. Spleen sections of the control and NPs treated groups show normal histological features which comprised ideal red and white pulp region. Kidney sections show normal-looking medulla and cortex in all four groups. Lung and heart sections of the control and NPs treated groups show normal histological features.

doses of NPs did not show any kind of increment in the count of different types of white blood cells (WBCs) i.e., neutrophils, lymphocytes, and monocytes in blood samples. These results also confirm that NPs are not creating an inflammatory behavior in the physiological milieu. So, it can be concluded that $Ch-Mn_3O_4$ NPs are safe and biocompatible in the physiological milieu.

## Discussion

This study was designed to determine whether an orally administrable novel nano-drug, $Ch-Mn_3O_4$ NPs, could be used as a redox medicine for the treatment of severe ulcerative colitis, a crucial clinical question considering the growing prevalence of the disease and non-availability of safe and effective therapeutic strategy. Our results provide direct evidence that treatment with the nanodrug ($0.25\,mg\,kg^{-1}$ BW, daily for 15 days) ameliorates both clinical symptoms (e.g., weight loss, shortening of colon length, rectal bleeding, inconsistent stool, DAI) and histological damages (e.g., intestinal epithelial barrier disruption, goblet cell depletion, granulocyte infiltration, crypt cell damage) of acute ulcerative colitis in DSS-intoxicated BALB/c mice. The extensive restoration of regular intestinal histological features (i.e., 'histological healing') accomplished through $Ch-Mn_3O_4$ NP treatment is immensely important in the clinical management of ulcerative colitis as available data indicate that histologic healing is associated with better outcomes in ulcerative colitis, including lower relapse rates and cancer risk[63]. Interestingly, most of the currently available treatment options are incapable of instigating histological healing. As a result, mucosal healing, routinely assessed by endoscopy, has emerged as the major therapeutic target though microscopic evidence of inflammation is common even in patients with clinically and sigmoidoscopically quiescent colitis[64], suggesting that endoscopy may underestimate the extent of the disease and possibly alter the clinical follow-up scheme[65]. Thus, the development of new-age therapies like the one investigated in this study will not only improve the treatment outcome but also support the emerging argument about raising the bar of drug activity assessment by implementing histologic remission as the ultimate treatment goal in clinical practice[63,65].

The favorable therapeutic effects illustrated by $Ch-Mn_3O_4$ NPs may perhaps be stemmed from suppressed local inflammation and maintenance of cellular redox equilibrium as manifested by decreased proinflammatory cytokines and reduced mediators of oxidative damage. The reduction in intestinal inflammation due to $Ch-Mn_3O_4$ NP treatment is consistent with our previous results[66] where a kidney-targeted different ligand functionalized manganese oxide nanoparticle reduced renal inflammation and cured chronic kidney disease. This study also provides a piece of direct evidence that $Ch-Mn_3O_4$ NPs can scavenge ROS, particularly $H_2O_2$, the longest-living one in the physiological milieu. Furthermore, it can generate ROS depending on the microenvironment, in turn, functions as a redox buffer. Considering the inseparable correlation between uncontrolled, persistent oxidative stress and inflammation, we postulate that the redox buffering activity of $Ch-Mn_3O_4$ NPs is the underlying reason behind the observed anti-inflammatory activity. This also supports our histological finding that $Ch-Mn_3O_4$ NPs prevent damage to the intestinal epithelium and restores regular colon architecture as several external and internal stimuli induce the formation of ROS in the gut, resulting in lipid peroxidation and apoptosis and colonic mucosal damage.

Our observation that prevention of mitochondrial damage from oxidative distress is one of the key molecular events in the therapeutic mechanism of $Ch-Mn_3O_4$ NPs provides proof-of-concept in support of the emerging hypothesis that mitochondrial redox homeostasis could be a putative drug target for the prevention of inflammation-associated disorders like ulcerative colitis[67]. It is well known that mitochondria are coordinators of cellular homeostasis via their role in energy production and oxidative metabolism, induction of apoptosis, regulation of calcium, production of ROS, and regulation of signal transduction and epigenomic intermediates[68]. In the intestine, mitochondrial metabolism and function play key roles in immune cell activation, intestinal epithelial cell barrier integrity and differentiation programs, and stemness[69]. Several prior studies have revealed the involvement of mitochondrial stress in the pathophysiology of ulcerative colitis but whether this is a cause or consequence of the disease is not known[70]. While mitochondria function as a platform for pro-inflammatory signaling cascade through activation of NF-κβ pathway[60], the pro-inflammatory mediators also alter mitochondrial function[62]. Interestingly, both processes increase mitochondrial oxidative distress or in other words dysregulate mitochondrial redox homeostasis, promoting a vicious inflammatory cycle. Furthermore, damage-associated molecular patterns (DAMPs) derived from mitochondria lead to NRLP3 inflammasome formation and caspase-1 activation, the processing of pro-IL-1β and pro-IL-18, and pyroptotic cell death[60,62]. In this study, the observed increase in mtROS, opening of mPTP, deregulation of mitochondrial membrane potential, and depletion in ATP levels due to disease pathogenesis functioned as mitochondrial stress response leading to activation of NF-κβ pathway and induction of multiple pro-inflammatory genes and interleukins (ILs). Observed overexpression of NF-κβ in lamina propria and increased IL-1β and IL-12 levels further support this argument. The DAMPs released from mitochondria due to loss of membrane integrity activated the NRLP3 inflammasome, thereby driving caspase-1 activation, also illustrated in the immunofluorescence study. On the other hand, $Ch-Mn_3O_4$ NPs regulated the cellular and mitochondrial ROS to homeostatic level, in turn, protected the structural and functional integrity of mitochondria (reflected in low mPTP, regular $\Delta\Psi_m$, and normal ATP content), thereby obstructing the causal vicious cycle of inflammation. The observed inhibition of mPTP opening by $Ch-Mn_3O_4$ NPs is a critical event in the therapeutic mechanism because recent studies have demonstrated a large pore formed by oligomerization of the voltage-dependent anion channel 1 (VDAC1), the mPTP formation mechanism, mediates DAMP movement across the outer mitochondrial membrane for immune cell activation, with a pathogenic role[71]. The down-regulation of pro-inflammatory cytokines further supports this argument. The findings that $Ch-Mn_3O_4$ NPs maintain cellular redox homeostasis and prevent pro-inflammatory signaling cascades through mito-protection corroborates our previous study that elucidated a similar redox-mediated nephroprotective mechanism by citrate-functionalized-$Mn_3O_4$ NPs in treatment of chronic kidney disease.

The origin of the catalytic mechanisms behind ROS generation by $Ch-Mn_3O_4$ NPs lies in the unique electronic structure of the nanodrug having closely packed tetrahedral and octahedral sites within a face-centered cubic oxygen framework. The valence band of the structure is dominated by tetrahedral $Mn^{2+}$, while the octahedral Mn-ions are in a Jahn-Tellar distorted, high-spin $Mn^{3+}$ state (also evident in the UV–vis spectra)[30,72]. The high-energy tetrahedral $Mn^{2+}$ spontaneously oxidizes to $Mn^{3+}$, followed by the removal of electrons from octahedral site, forming $Mn^{4+}$. When three electrons per formula unit are removed, the valence band becomes dominated by oxygen states, and electrons are extracted from $O^{2p}$ orbitals instead of tetrahedral $Mn^{3+}$, resulting in the formation of oxygen holes on the nanodrug surface. The holes generate transient free radicals upon interaction with dissolved molecular oxygen. On the other hand, when in abundance ROS are readily adsorbed by the nanodrug to replenish its initial electronic configuration.

The limitations of this study provide scope for future work. A comprehensive investigation of the molecular mechanism of action and its effect on other inflammatory pathways and mediators of ulcerative colitis would be beneficial to translate this work into clinical trial. The use of a genetically modified rodent model or extending it to the treatment of IBD could also provide further information. One huge advantage of this therapeutic strategy is the apparent non-toxicity of this nanodrug. However, a more detailed toxicity study particularly at the molecular level is important to meet the FDA endorsement requirements.

There are very few published articles in contemporary literature that report a promising alternative approach for the treatment of ROS-induced inflammatory diseases like ulcerative colitis. On the other hand, many new reports are confirming the involvement of redox imbalance behind the pathogenesis of ulcerative colitis. Our study suggests that Ch-Mn$_3$O$_4$ NPs could be an efficient redox medicine to attenuate colon injury and ulcer formation as evidenced by the reduction in clinical symptoms, restoration of normal colon length, decrease in the infiltration of immune cells, reduction in molecular markers of inflammation, down-regulated pro-inflammatory cytokines, and initiation of regenerative glands at the site of the ulcer. The molecular mechanism involves the redox buffering of cellular redox state by Ch-Mn$_3$O$_4$ NPs and protects the mitochondria from ROS damage. The findings strongly suggest the translational potential of Ch-Mn$_3$O$_4$ NPs as a redox nanoparticle for the treatment of UC.

## Methods

**Synthesis of nanoparticle (Ch-Mn$_3$O$_4$).** A reported method was followed for the template-free synthesis of Mn$_3$O$_4$ in ambient temperature and pressure[23,73]. For the functionalization of nanoparticles, a chitosan solution was prepared using 1% acetic acid. Then the pH was adjusted to 6 and prepared Mn$_3$O$_4$ nanoparticle was added to the ligand solution. The mixture was then kept for stirring for 36 h. After 36 h, using a syringe filter (diameter 0.22 μm) successfully capped NP was separated from non-functionalized bigger nanoparticles.

**Characterization tools and techniques.** Images of HRTEM and transmission electron microscope (TEM) were obtained using an FEI TecnaiTF-20 field emission HRTEM operating at 200 kV. Samples for the TEM analysis were prepared by drop-casting the nanoparticle solution on a 300-mesh amorphous carbon-coated copper grid and for drying propose it was kept overnight at ambient temperature. Using Shimadzu UV–vis 2600 spectrometer absorbance spectra of chitosan and chitosan-capped Mn$_3$O$_4$ (Ch-Mn$_3$O$_4$) NPs were measured. Fluorescence excitation and emission spectra of the prepared nanoparticle were accessed using fluorolog, Model LFI-3751 (Horiba-Jobin Yvon, Edison, NJ) spectrofluorimeter.

**Quantification and characterization of ROS generated by Ch-Mn$_3$O$_4$ NPs.** DCFH a very popular chemical for quantification of ROS generation was prepared from DCFH-diacetate via de-esterification reaction at room temperature following a standardized protocol described in earlier studies[23,32,74]. ROS generated in the reaction medium reacted with non-fluorescent DCFH and convert to fluorescent DCFH which has a characteristic fluorescence emission maxima at 520 nm when excited at 488 nm. DCFH was allowed to react with increasing concentration of Ch-Mn$_3$O$_4$ NPs in an aqueous medium, and fluorescence was recorded using fluorolog, Model LFI-3751 (Horiba-JobinYvon, Edison, NJ) spectrofluorimeter. Chemiluminescence produced by oxidized luminol was recorded using fluorolog, Model LFI-3751 (Horiba-JobinYvon, Edison, NJ)[23].

**The ABTS radical scavenging assay.** The assay was performed using the standard kit purchased from Sigma-Aldrich. We have used 0.5 μg ml$^{-1}$ NPs for the experiment. The kit provided trolox solutions as standard. A standard curve for the Trolox solution was done using the provided solutions. After 5 min of incubation, we observed the absorbance at 405 nm using Shimadzu UV–vis 2600 spectrometer. The anti-oxidant potential was calculated using the following equation:

$$\text{Anti} - \text{oxidant potential (mM)} = \frac{\text{Absorbance } at\ 405 - intercept}{\text{Slope}} \times \text{Dilution factor}$$

(1)

**Culture of human embryonic kidney cells (HEK 293)[75].** HEK 293 cells were maintained at 37 °C in 5% CO$_2$ in RPMI 1640 growth medium (Himedia, India)

that contained 10% fetal bovine serum (Invitrogen, USA), L-glutamine (2 mM), penicillin (100 units mL$^{-1}$), and streptomycin (100 ng mL$^{-1}$) (Sigma, USA). Before experimentation, the cells were washed twice and incubated with RPMI 1640 medium (FBS, 0.5%) for 1 h and then treated as described in the figure legends.

**Measurement of cell viability.** Cell viability was assessed by MTT. All cell lines were plated in 96-well plates at a density of $1 \times 10^3$ cells/well and cultured overnight at 37 °C. The treatments were performed as described in the figure legends. Next, MTT (5 mg ml$^{-1}$; Himedia, India) was added to each well, with a final concentration of ~0.5 mg ml$^{-1}$, and the cells were cultured for 4 h at 37 °C in a 5% CO$_2$ atmosphere. The resultant purple formazan was dissolved by the addition of 10% sodium dodecyl sulfate (Sigma, USA) and the absorbance was read at 570 and 630 nm using a microplate reader (BioTek, USA).

**Rodent model.** All animal studies and experimental procedures were performed at the Central Animal Facility, Department of Zoology, Uluberia College, India (Reg. No. 2057/GO/ReRcBi/S/19/CPCSEA) following the protocol approved by the Institutional Animal Ethics Committee (Ethical Clearance No. 03/S/UC-IAEC/01/2019) as per standard guideline of Committee for the Control and Supervision of Experiments on Animals (CCSEA), Ministry of Fisheries, Animal Husbandry and Dairying, Government of India. Healthy non-diabetic BALB/c mice of both sexes (age: 5–6 weeks, body weight, BW: 20 ± 2.3 g) were used in the current study. All the experimental animals were kept in standard, clean polypropylene cages and maintained normal environmental conditions as per guidelines (temperature 21 ± 1 °C; relative humidity 45–55%; 1:1 light and dark cycle). Pathogen-free water and a standard laboratory pellet diet for mice (Saha Enterprise, Kolkata, India) were available ad libitum throughout the experimental period. All mice were allowed to acclimatize for 2 weeks before the treatment. Autoclaved nest material and paper houses served as cage enrichment for mice. All the mice were allowed to acclimatize for 2 weeks before beginning the treatment protocol. Animal cages were always randomly assigned to treatment or control groups.

**Study design.** The purpose of this study was to assess the potential of orally-administrable Chitosan-capped Mn$_3$O$_4$ NPs (Ch-Mn$_3$O$_4$ NPs) as a therapeutic agent for rapid recovery from ulcerative colitis. To define in vivo effects of Ch-Mn$_3$O$_4$ NPs, we selected BALB/c mice as our preclinical animal model. In brief, the therapeutic potential of Ch-Mn$_3$O$_4$ NPs was evaluated along with histopathological changes, inflammatory cytokines level, mitochondrial activity, etc. And DAI was monitored in a time-dependent manner. The experimental model of UC was established according to the previously stated protocol[42].

Mice were randomly divided into four groups (N = 10/group): (1) control; (2) DSS; (3) DSS+Ch-Mn$_3$O$_4$ NPs; (4) Ch-Mn$_3$O$_4$ NPs control. Animals of Group 1 served as control and received normal saline (150 μL; oral). Animals of Group 2 served as diseased model and received 5% DSS in drinking water for 12 days. Animals of Group 3 received 5% DSS in drinking water for 12 days. Animals of Group 4 served as NPs control and received single oral dose of Ch-Mn$_3$O$_4$ NPs (0.25 mg kg$^{-1}$ BW) on a daily basis for 15 days. After development of severe UC on day 13, Group 2 animals left untreated, Group 3 animals received single oral dose of Ch-Mn$_3$O$_4$ NPs (0.25 mg kg$^{-1}$ BW) on a daily basis for 15 days. All doses were finalized based on reported literature,bio-compatibility study andpilot experimentation. As chitosan treatment did not improve the symptoms of ulcerative colitis, data were not included in the manuscript.

**Experimental design of sub-chronic toxicity study.** Mice were randomly divided into four groups (N = 10/group). Ch-Mn$_3$O$_4$ NPs was administrated orally at three dose levels i.e., 0.25 mg kg$^{-1}$ BW, 0.5 mg kg$^{-1}$ BW, and 1 mg kg$^{-1}$ BW corresponding to low, intermediate, and high dose respectively for 28 days. Normal saline was administrated to the mice of the control group. Treatment was done once daily for 28 days.

**Evaluation of DAI score[43].** Body weights, hemoccult, gross blood, and stool consistency were analyzed on a daily basis. DAI was calculated by scoring the percentage of weight loss, intestinal bleeding [no blood, occult blood (haemoccult +), or gross blood], and stool consistency (normal stool, loose stool, or diarrhea), as previously described[43]. The clinical features were scored separately and then correlated with a histological score: DAI = (body weight loss score) + (diarrhea score) + (rectal bleeding score) (Fig. 2a).

**Blood collection[76].** At the end of the experimental period, the animals were euthanized and decapitated after being fasted. Blood was collected from the retro-orbital plexus just before the sacrifice, and kept in sterile non-heparinized tubes for further analysis.

**Serum isolation.** After the collection of blood, kept in sterile non-heparinized tubes in a slanting position at a 45° angle for 45 min and centrifuged at 3500×g for 20 min. The clear serum (straw color) was obtained and used in subsequent biochemical analysis.

**Measurement of inflammatory cytokines level**. All serum samples were sterile, hemolysis-free, and stored at −20 °C before the determination of the biochemical parameters. The concentration of inflammatory cytokines (IL-1β and IL-12) was measured according to the respective protocol provided by kit manufacturers (G-Biosciences, St. Louis, MO, USA).

**Assessment of hematological parameters**. For hematological studies, the blood was collected in heparinized tubes. Blood cell count was done using blood smears in Sysmax-K1000 Cell Counter. The parameters studied were hemoglobin, total red blood cells, hematocrit, packed cell volume, mean corpuscular volume (MCV), mean corpuscular hemoglobin (MCH), MCH concentration, platelets, total WBCs, etc.

**Occult blood test[77]**. For a collection of stool samples, the individual mouse was placed in an empty autoclaved cage without bedding. Then the mouse was allowed to defecate normally. Once the mouse has defecated 2–3 fecal pellets, the mouse was removed from the collection cage. Now the fecal pellets were collected in a sterilized centrifuge tube using a sterile toothpick. And the samples were stored in a −80 °C freezer. The two fields occult blood test in stool samples was performed using commercially available kits (HEMOSPOT; Coral Clinical Systems, Goa, India) following the protocol described by respective manufacturers.

**Histological examination**. At the end of the experimental period, the animals were sacrificed and colons were collected for histological examination[78]. For microscopic evaluation, a conventional technique of paraffin wax sectioning and differential staining was used. Tissues were fixed with 4% paraformaldehyde, embedded in paraffin, and cut into a 5 μm thick section. After de-waxing and gradual hydration with ethanol (Merck, USA), the sections were stained with hematoxylin and eosin (SRL, India). The sections were then observed under an optical microscope (Olympus, Tokyo, Japan). It is noteworthy to mention here that the histopathologist was blinded to the treatment groups while scoring and evaluating the samples. For immunohistochemistry, colon sections were incubated for 60 min with rat anti-mouse caspase-1, NF-κβ p-65 antibodies (Santa Cruz Biotechnology) followed by a 30 min incubation with 10 mg ml−1 FITC conjugated mouse IgGFc binding antibody (Santa Cruz Biotechnology). To check CD-68 expression, colon sections were incubated for 60 min with rat anti-mouse CD-68 antibody (Santa Cruz Biotechnology, India) followed by a 30 min incubation with 10 mg ml−1 HRP conjugated rabbit anti-rat secondary antibody (Santa Cruz Biotechnology, India). After the detection of peroxidase activity with 3-amino-9-ethylcarbazole (Sigma, USA), sections were counterstained with Mayer's hematoxylin (SRL, India). After that the tissues were viewed with Leica DM6B microscope.

**Histology scoring**. The Geobes score has been used for histopathological analysis to evaluate all aspects of mucosal injury in ulcerative colitis. The scoring system is divided into 6 grades: architectural changes [grade 0], chronic inflammatory infiltrate [grade 1], lamina propria neutrophils and eosinophils [grade 2], neutrophils in the epithelium [grade 3], crypt destruction [grade 4] and erosions or ulcerations [grade 5], and each grade of the score is divided in 4 subcategories[46]. It is noteworthy to mention here that the histopathologist was blinded to the treatment groups while scoring.

**Mitochondria isolation and mitochondrial function determination**. Mitochondria were isolated from the mouse colon following the method reported earlier with slight modification[79]. In brief, colons were excised and homogenized in a tissue homogenization medium containing 225mMD-mannitol, 75 mM sucrose, 0.05 mM EDTA, 10 mM KCl, and 10 mM HEPES (pH 7.4). The homogenates were centrifuged at $600 \times g$ for 15 min and the resulting supernatants were centrifuged at $8500 \times g$ for 10 min. The pellets were washed thrice and resuspended in the same buffer. All procedures were performed at 4 °C. Mitochondrial function was evaluated by determining $\Delta\Psi_m$ using JC-1 (Sigma, MO, USA), and ATP production (Abcam, Germany). And them mPTP opening was evaluated in terms of mitochondrial swelling by monitoring the decrease in absorbance at 540 nm after the addition of $CaCl_2$ (100 mM).

**Bio-distribution**. For the bio-distribution study, all the organs of the experimental mice were collected after 7 days of daily single-dose treatment with Ch-Mn₃O₄ NPs. After collection, all the tissue samples were dried immediately using liquid nitrogen and weighed. The detection and quantification of Manganese (Mn) content in tissue samples was carried out using inductively coupled plasma optical emission spectroscopy (ICP-OES) (Agilent Technologies, USA). The tissue samples were digested following the USEPA methods 200.2 revision 2.8 with slight modifications for performing the ICP–OES method. After proper filtration, the digested solution was diluted with deionized water to a final volume of 10 mL.

**Statistic and reproducibility**. All quantitative data are expressed as mean ± standard deviation unless otherwise stated. One-way analysis of variance (ANOVA) followed by correction of false discovery rate (post hoc FDR: two-stage step-up method of Benjamini, Krieger, and Yekutieli) for multiple comparisons was performed for comparison between multiple groups[80]. Beforehand, the normality of each parameter was checked by normal quantile–quantile plots. Sample size in our animal studies was determined following the standard sample sizes previously used in similar experiments as per relevant literature. Designated sample size (in figure legends) always refers to biological replicates (independent animals). GraphPad Prism v8.0 (GraphPad Software), and Sigmaplot v14.0 (Systat Software, Inc.) were used for statistical analysis. For all comparisons, a $P$ value $< 0.05$ was considered to be statistically significant.

**Reporting summary**. Further information on research design is available in the Nature Portfolio Reporting Summary linked to this article.

## Data availability
The authors confirm that all data are available in the main text and supplemental information. The source data behind the graphs can be found in Supplementary Data 1. Any other related data that support the findings of this study are available from the corresponding author, S.K.P. (skpal@bose.res.in), upon reasonable request.

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

## Acknowledgements

The authors would like to acknowledge the Deanship of Scientific Research at Umm Al-Qura University, for supporting this work by Grant code: 23UQU4280406DSR001. Dr. Ziad Moussa is grateful to the United Arab Emirates University (UAEU) and to the Research Office for supporting the research developed in his laboratory and reported herein (SUREPLUS Grant code G00003918).

## Author contributions

S.M., S.A.A., and S.K.P. designed the experiments. S.M., A.A., R.G., and M.S. did synthesis, characterization, and in vitro studies. S.M., M.D., and S.S.B. performed cellular studies. S.M., A.A., S.D., and M.D. conducted animal experiments. A.K.D. performed the histological studies and analysis. S.M., A.A., D.P., D.B., A.S.A.A., A.K.M., M.M.A., Z.M., S.A.A., and S.K.P. discussed and analyzed the results. S.M. wrote the paper and all authors contributed toward the writing of the final version.

## Competing interests

The authors declare no competing interests.
