## [Peer Review File · Communications Biology]

Reviewers' comments:

Reviewer #1 (Remarks to the Author):

Authors developed Chitosan Functionalized Mn₃O₄ (Ch- Mn₃O₄ NPs) to regulate the ROS in colitis mice. Basically, the manuscript was well-written, and therapeutic treatment of Ch- Mn₃O₄ NPs was impressively observed in the DSS-induced colitis mice at low dose. However, the novelty and several important evaluations are not well investigated. The Biodistribution of Ch- Mn₃O₄ NPs in GI tract and its mechanism on counteracting with ROS should be investigated. Therefore, current manuscript should not be considered for publication at this version. Following comments should be addressed:

- In the Introduction section, Author should introduce various antioxidant nanomedicines have been investigated to regulate redox equilibrium in colitis mice.

- Why the stability of NPs can be monitored by measuring the absorbance at 280 nm? The size zeta potential and aggregation should be evaluated as function of time to confirm the stability under different pH values (at GI tract)

- The size of NPs was quite small (< 10 nm). The Biodistribution of orally administered NPs should be performed to understand the distribution of NPs in blood, organ, especially the accumulation in GI tract (colon). This is one of important investigation when using oral NPs

- It is not clear on the activity of NPs against ROS by DCF assay. The ROS scavenging of NPs should be evaluated with various methods

- In the DSS-induced colitis model, most reports showed that 7 days and 3-5% DSS were popular protocol to induce colitis in mice; longer DSS treatment might lead to mortality. Please carefully describe the treatment of DSS and NPs. DSS 28 days and NPs 14 days, is that correct?

- Reduction of Mortality rate was not investigated as author mentioned in the abstract? Please check

- Figure 4c, the error bar should be added and statistically analyzed

- The histological image (Figure 3), the DSS+ Ch-Mn₃O₄ NPs looks not different from DSS group although the histological score was significantly different. Please check and revise

Reviewer #2 (Remarks to the Author):

The manuscript describes a very good, successful, innovative study on colonic delivery of metallic nanoparticles for UC treatment. The writing is very good overall. However, the problem is what is missing. Several minor things, as it turns out. Fixing these small concerns will result in a very good manuscript.

Line 1: I suggest using 'BY SCAVENGING MUCOSAL ROS' instead of 'through Modulation of Exposome'
Line 11 and 16: It is better to use the word 'nanoparticle' instead of 'nanomedice' and in the whole manuscript.

Line 17 and 19: Your abstract's conclusion looks unclear, please re-write it again according to my suggested title.

Line 20: Please don't repeat the same word in the main title and keywords. Change all repeated words.

Line 28: Use the full name 'ulcerative colitis' and its abbreviation 'UC' and then, when repeated use 'UC'.

Line 39: Please add other treatment options for UC, 5-ASA, corticosteroids and immunosuppressants.

Line 81: The authors should also add this similar article as a reference: Yuan Cheng, Chaoqun Cheng, Jia Yao, Yijun Yu, Yufeng Liu, He Zhang, Leiying Mia and Hui. Mn₃O₄ Nanozyme for Inflammatory Bowel Disease Therapy, *Advanced Therapeutics*, 2021, 2100081:1-9.

Line 92: The authors should also add this article to proof that chitosan was successfully used to target colitis: xxxxoshi

Line 123: Did the authors check the stability of their NPs in different pH (2.1 for stomach, 6.8 for small intestine and 7.4 for colon) similar to GIT conditions?

Line 146: The author should write the full name of DCFH.

Lines 423: The authors should add the reference that they follow to prepare their nanoparticle.

Lines 446, 437, 493 and 499, 515: The authors should add the references.

Reviewer #3 (Remarks to the Author):

This is a routine study. Physiological observations including fecal blood and defecation frequency, as well as elevated inflammatory molecular markers, established inflammatory UC in a mouse model after dextran sulfate (DSS) induction. The application of this nanomaterial greatly reduces ROS in the foci, and reduces the mortality of the disease, as well as the inflammation of the disease, the manuscript reflects many works, but the significance is not significant.

General comments:

1. Material characterization and cytotoxicity hemolysis assays need to be supplemented.
2. Lack of basic experiments and interventions to confirm the reliability of the experiments.
3. Lack of confirmation of the effects of modified materials on other organs.
4. Application of nanomaterials The timing and trend of specific anti-inflammatory oxidative stress have not been identified.
5. It is recommended to add some western blotting and PCR to enhance the reliability of the experiment.

Other comments:

1. The quality of the numbers is hard to read, there are no numbers.
2. There are many grammatical errors in the manuscript. English should be edited.
3. It is recommended to change to other magazines.

सत्येन्द्र नाथ बसु राष्ट्रीय मौलिक बिज्ञान केन्द्र
**SATYENDRA NATH BOSE NATIONAL
CENTRE FOR BASIC SCIENCES**

Reviewer:1

Recommendation: Major revisions needed as noted.

Comments:

Authors developed Chitosan Functionalized Mn_3O_4 (Ch- Mn_3O_4 NPs) to regulate the ROS in colitis mice. Basically, the manuscript was well-written, and therapeutic treatment of Ch- Mn_3O_4 NPs was impressively observed in the DSS-induced colitis mice at low dose. However, the novelty and several important evaluations are not well investigated. The Biodistribution of Ch- Mn_3O_4 NPs in GI tract and its mechanism on counteracting with ROS should be investigated. Therefore, current manuscript should not be considered for publication at this version. Following comments should be addressed:

Reply: We would like to thank the learned reviewer for carefully reading the manuscript and appreciating our work. We have addressed all the queries point by point in the following section.

Query 1: In the Introduction section, Author should introduce various antioxidant nanomedicines have been investigated to regulate redox equilibrium in colitis mice.

Reply 1: We would like to thank the learned reviewer for carefully reading our manuscript. We have incorporated a discussion on the available antioxidant nanomedicines that has been found to be effective against ulcerative colitis in preclinical mice model in the revised manuscript. (Page: 5 Line Number: 5-9).

Query 2: Why the stability of NPs can be monitored by measuring the absorbance at 280 nm? The size zeta potential and aggregation should be evaluated as function of time to confirm the stability under different pH values (at GI tract)

Reply 2: We thank the learned reviewer for his valuable suggestion. It has to be noted that, the synthesized nanoparticle has its characteristic spectral signature at 280 nm. By monitoring the absorbance of the aqueous solution of the nanoparticle at 280 nm for a longer period we can determine its aqueous stability. Any change in the absorbance indicates degradation or precipitation of the NPs.

As suggested by the learned reviewer we have check the aqueous stability of the nanomedicine at different pH by monitoring its absorbance at 280 nm (Supplementary Figure S3a).

The size and the zeta potential under different pH values has also been investigated. The data are included for the learned reviewer (Supplementary Figure S3b).

BLOCK-JD, SECTOR-III, SALT LAKE, KOLKATA-700098

Phones- 0091-(0)33-2335 5705-8, 2335 3057/61, 2335 0312/1313

E-mail: root@bose.res.in

Webpage: <http://www.bose.res.in>

Fax: 0091-(0)33-2335 3477

FUNDED BY THE DEPARTMENT OF SCIENCE & TECHNOLOGY, GOVT. OF INDIA

सत्येन्द्र नाथ बसु राष्ट्रीय मौलिक बिज्ञान केन्द्र SATYENDRA NATH BOSE NATIONAL CENTRE FOR BASIC SCIENCES

and the zeta potential under different pH values has also been investigated. The data are included for the learned reviewer.

It has to be noted that the structural stability of the NPs in different pH remains same. Our TEM studies in different pH confirm that there is no significant effect of pH in shape and size of the core nanoparticle (Supplementary Figure S3c-3f). The aqueous stability of the NP in different pH have also been studied. From the absorbance and zeta potential studies we can conclude that the synthesized NP is precipitating out from the solution at higher pH as chitosan molecules tends to aggregate at higher pH. So, this characteristic of our NP is very much useful in intestine targeted delivery which was the aim of this study. The issue of stability in different pH has been included in the revised manuscript. (Page: 8, Line Number: 3-14)

Query 3: The size of NPs was quite small (< 10 nm). The Biodistribution of orally administered NPs should be performed to understand the distribution of NPs in blood, organ, especially the accumulation in GI tract (colon). This is one of important investigation when using oral NPs.

Reply 3: We appreciate the concern raised by the learned reviewer. We have checked the bio-distribution of the NPs in different organs using ICPOES. But there was no significant accumulation of NPs in any organs other than colon. We have included the data for colon in the figure (Figure 6c) and discussed the issue in the revised manuscript. (Figure 6C, Page: 18 Line Number: 5-11; Page: 29, Line Number: 21-24, Page: 30, Line number: 1-4)

We have already performed the sub-chronic toxicity study in preclinical mice following the well-recognized testing protocol of OECD. No signs of toxicity or abnormalities were found in any of the organ specific studied. Further, we have already added this toxicity study in our revised manuscript. (Figure 6d, Page:18, Line:12- Page:19, Line:1-2)

Query 4: It is not clear on the activity of NPs against ROS by DCF assay. The ROS scavenging of NPs should be evaluated with various methods.

Reply 4: We appreciate the concern raised by the learned reviewer. The DCF assay, only with Ch-Mn₃O₄ NPs, was performed to show the ROS generation activity of the NPs. The next DCF assay with both Ch-Mn₃O₄ NPs and a known ROS (H₂O₂) was performed to establish the ROS scavenging as well as the redox buffering property of the NPs. However, as suggested by the learned reviewer we have again performed ABTS assay to confirm the anti-oxidant property of the NPs (Supplementary Figure S4b). The issues has been addressed in the revised manuscript. (Page: 9, Line Number: 1-7; Page: 25, Line Number: 3-8)

सत्येन्द्र नाथ बसु राष्ट्रीय मौलिक बिज्ञान केन्द्र
**SATYENDRA NATH BOSE NATIONAL
CENTRE FOR BASIC SCIENCES**

Query 5: In the DSS-induced colitis model, most reports showed that 7 days and 3-5% DSS were popular protocol to induce colitis in mice; longer DSS treatment might lead to mortality. Please carefully describe the treatment of DSS and NPs. DSS 28 days and NPs 14 days, is that correct?

Reply 5: We thank the learned reviewer for the carefully reading of our manuscript. We have induced the colitis model in mice by administering 5% DSS orally for 12 days. From the 13th day onwards, the NP were administered for the next 15 days. Please note that no DSS was given to the mice during the treatment period.

Query 6: Reduction of Mortality rate was not investigated as author mentioned in the abstract? Please check.

Reply 6: We appreciate the concern of the learned reviewer. We investigated the mortality rate however, data was not shown. The survivability percentage of experimental animals was monitored throughout the study regime. After the treatment period, we have found 60% of the experimental animals survived in our DSS treated group. On the other hand there was 90% survivability percentage in DSS+ Ch-Mn₃O₄ NPs treated group. Hence, treatment with the Ch-Mn₃O₄ NPs successfully prevented the mortality of animals suffering from ulcerative colitis in our study. (Supplementary Figure S8) (Page: 11, Line Number: 10-11)

Query 7: Figure 4c, the error bar should be added and statistically analysed.

Reply 7: We thank the learned reviewer for his kind suggestion. We have added the error bar in the Figure 4c and also include the statistical analysis in the revised version. (Page: 14, Line Number: 18-20, 22-23, Page: 15, Line Number: 1, 5-6, 7-9)

Query 8: The histological image (Figure 3), the DSS+ Ch-Mn₃O₄ NPs looks not different from DSS group although the histological score was significantly different. Please check and revise.

Reply 8: We acknowledge the concern of the learned reviewer regarding the histopathological analysis and histology scoring of H and E stained colon tissue sections of experimental mice. And we also thank the learned reviewer for the valuable suggestions in this context. As suggested by the reviewer, we performed the re-verification of the H and E stained histology slides especially DSS and DSS+Ch-Mn₃O₄ NPs groups under the supervision of Prof. (Dr.) Anjan Kumar Das, who is a renowned professor and practicing clinical histopathologist in a Govt. medical college and hospital. He is also one of the co-authors of this manuscript.

After rigorous observation Dr. Das confirmed that the histological features of the images of "DSS and DSS + Ch-Mn₃O₄ NPs groups" are different from each other in the context of histopathological manifestation of ulcerative colitis disease. Dr. Das also reported that

BLOCK-JD, SECTOR-III, SALT LAKE, KOLKATA-700098

Phones- 0091-(0)33-2335 5705-8, 2335 3057/61, 2335 0312/1313

E-mail: root@bose.res.in

Webpage: <http://www.bose.res.in>

Fax: 0091-(0)33-2335 3477

FUNDED BY THE DEPARTMENT OF SCIENCE & TECHNOLOGY, GOVT. OF INDIA

सत्येन्द्र नाथ बसु राष्ट्रीय मौलिक बिज्ञान केन्द्र SATYENDRA NATH BOSE NATIONAL CENTRE FOR BASIC SCIENCES

the histopathology report for all the slides of four groups is accurate and same as our previous observation.

The H and E stained colon tissue sections of the “DSS group” (Ulcerative colitis model cohort) shows the typical histopathological manifestation of severe ulcerative colitis disease. It shows well demarcated superficial mucosal ulceration with fairly dense mixed inflammatory cells infiltration within lamina propria. In addition, presence of severe cryptitis and crypt abscesses, moderate increase in the number of neutrophils in lamina propria along with the presence of small areas of granulation tissues and mucosal fibrosis in lamina propria indicate towards severe architectural abnormalities in colon. (Fig. 3b, Supplementary Fig. S7b)

On the other hand the H and E stained colon tissue sections of the “DSS + Ch-Mn₃O₄ NPs group” shows healing and remarkable improvements in the histopathological features of the damaged colon. It shows healed ulcerated area by regenerating epithelium along with several regenerative glands. The extent of inflammatory cells infiltration reduced significantly along with normal looking crypt cells, usual number of neutrophils in lamina propria and epithelium, and regenerating epithelial layer with negligible level of adjacent inflammation. In brief, it shows almost healthy mucosal and submucosal layer, muscularis propria and serosa. (Fig. 3c, Supplementary Fig. S7c)

In addition to that the H & E stained colon tissue sections isolated from both control and Ch-Mn₃O₄ NPs control cohorts show normal histological features of colon like healthy mucosal and submucosal layer, typical muscularis propria and serosa [Fig. 3a and Fig.3d, Supplementary Fig. S7a, S7d]

In the revised manuscript, low magnification histological images of all the study groups have also been added for clear understanding (Supplementary Fig. S7). Additionally we have marked the architectural abnormalities, ulceration, infiltration of inflammatory cells, distorted crypts and crypt abscess in histological images of “DSS group” (Fig. 3b, Supplementary Fig. S7b). We have also pointed out the healed ulcer area, presence of regenerative glands and normal crypts in respective images of “DSS + Ch-Mn₃O₄ NPs group” (Fig. 3c, Supplementary Fig. S7c).

Secondly as suggested by the learned reviewer, Dr. Das has also evaluated the histology score of all the respective study groups again. He has prepared the histology score of H and E stained colon tissue sections according to the original “Geboes Score”, the most commonly used histological score to evaluate all aspects of mucosal injury in ulcerative colitis disease. The scoring pattern is divided in 6 grades i.e. architectural changes [grade 0], chronic inflammatory infiltrate [grade 1], lamina propria eosinophils and neutrophils and [grade 2: 2A and 2B], neutrophils in epithelium [grade 3], crypt destruction [grade 4] and erosions or ulcerations [grade 5], and each grade of the score is divided in 4 subcategories to evaluate the extent of severity. Here severity of the disease especially the histological abnormality is proportional to the histology score. The

सत्येन्द्र नाथ बसु राष्ट्रीय मौलिक बिज्ञान केन्द्र
**SATYENDRA NATH BOSE NATIONAL
CENTRE FOR BASIC SCIENCES**

details of the scoring system of Geboes Score has been added under the materials and method section in the revised manuscript. (Page: 29, Line Number: 4-10)

Dr. Das has confirmed that the histological score of all the groups is correct and same as mentioned in the manuscript already. The Geboes Score for ulcerative colitis model cohort (DSS group) was significantly high indicating severity of colonic damage i.e. architectural abnormalities, chronic inflammatory cells infiltration, presence of neutrophils in lamina propria and epithelium, destruction of the crypts and presence of ulcer and granulation tissue (Geboes Score: 14.00 ± 0.81 vs 0.10 ± 0.31 of control cohort, $p < 0.0001$, one-way ANOVA, $F(3, 22.10) = 844.7$). [Supplementary Fig. S6 and S7]. Treatment with Ch-Mn₃O₄ NPs decreased the score within the normal range which is the confirmation of the histological healing of the colonic lesions, reduced inflammation and restoration of architectural integrity [Supplementary Fig.S6 and S7]. The Geboes Score for control and Ch-Mn₃O₄ NPs control cohorts were similar [Supplementary Fig.S6 and S7]. As suggested by the honourable reviewer, histology scoring and detailed explanation have been added in the revised manuscript. Additionally the low magnification histological images have been added and Geboes scoring grade wise histopathological features have also been marked in respective histological images of all the groups (Fig. 3C, Supplementary Fig. S7, Page: 11-13)

Dr. Das's specimen study notes and detailed histology scoring report during observation of the slides based on which we wrote the manuscript have been attached with this letter for your kind perusal.

Reviewer: 2

Recommendation: Not appropriate for ACS Applied Materials & Interfaces.

Comments:

The manuscript describes a very good, successful, innovative study on colonic delivery of metallic nanoparticles for UC treatment. The writing is very good overall. However, the problem is what is missing. Several minor things, as it turns out. Fixing these small concerns will result in a very good manuscript.

Reply: We would like to thank the learned reviewer for carefully reading the manuscript and appreciating our work. We have addressed all the queries point by point in the following section.

Query 1: Line 1: I suggest using 'BY SCAVENGING MUCOSAL ROS' instead of 'through Modulation of Exposome'.

Reply 1: We appreciate the concern of the learned reviewer. We have changed the title of the manuscript and replaced the phrase 'through Modulation of Exposome' with

BLOCK-JD, SECTOR-III, SALT LAKE, KOLKATA-700098

Phones- 0091-(0)33-2335 5705-8, 2335 3057/61, 2335 0312/1313

E-mail: root@bose.res.in

Webpage: <http://www.bose.res.in>

Fax: 0091-(0)33-2335 3477

FUNDED BY THE DEPARTMENT OF SCIENCE & TECHNOLOGY, GOVT. OF INDIA

सत्येन्द्र नाथ बसु राष्ट्रीय मौलिक बिज्ञान केन्द्र
**SATYENDRA NATH BOSE NATIONAL
CENTRE FOR BASIC SCIENCES**

'through Modulation of Cellular Redox State'. It has to be noted that the nanoparticle exerts its therapeutic effect not only by scavenging the ROS of the medium. It actually balances or buffers the cellular redox state thereby helps the cell to maintain its redox homeostasis. To highlight this mechanism, we have mentioned the modulation of redox state in the title of the manuscript.

We shall be highly obliged if the learned reviewer grants us the permission to keep this revised title. (Page:1 Line Number: 1-2)

Query 2: Line 11 and 16: It is better to use the word 'nanoparticle' instead of 'nanomedice' and in the whole manuscript.

Reply 2: We thank the learned reviewer for the valuable suggestion. We have changed the term nanomedicine to nanoparticle in the revised manuscript. (Page:1, Line:1; Page:2, Line:8,9,13,15; Page:23, Line:22)

Query 3: Line 17 and 19: Your abstract's conclusion looks unclear, please re-write it again according to my suggested title.

Reply 3: We thank the learned reviewer for the valuable suggestion. We have modified the abstract in revised manuscript as per your kind suggestion. (Page:2, Line: 14-17)

Query 4: Line 20: Please don't repeat the same word in the main title and keywords. Change all repeated words.

Reply 4: We appreciate the concern raised by the learned reviewer. We have changed the keywords in the revised version. (Page:2, Line:18-19)

Query 5: Line 28: Use the full name 'ulcerative colitis' and its abbreviation 'UC' and then, when repeated use 'UC'.

Reply 4: We thank the learned reviewer for the suggestion. We have addressed the issue in the revised manuscript. (Page: 2, Line:6)

Query 6: Line 39: Please add other treatment options for UC, 5-ASA, corticosteroids and immunosuppressants.

Reply 6: We thank the learned reviewer for the suggestion. We have addressed the issue in the revised manuscript. (Page: 3, Line: 11-21)

Query 7: Line 81: The authors should also add this similar article as a reference: Yuan Cheng, Chaoqun Cheng, Jia Yao, Yijun Yu, Yufeng Liu, He Zhang, Leiying Mia and Hui.

BLOCK-JD, SECTOR-III, SALT LAKE, KOLKATA-700098

Phones- 0091-(0)33-2335 5705-8, 2335 3057/61, 2335 0312/1313

E-mail: root@bose.res.in

Webpage: <http://www.bose.res.in>

Fax: 0091-(0)33-2335 3477

FUNDED BY THE DEPARTMENT OF SCIENCE & TECHNOLOGY, GOVT. OF INDIA

सत्येन्द्र नाथ बसु राष्ट्रीय मौलिक बिज्ञान केन्द्र
**SATYENDRA NATH BOSE NATIONAL
CENTRE FOR BASIC SCIENCES**

Mn3O4 Nanozyme for Inflammatory Bowel Disease Therapy, Advanced Therapeutics, 2021, 2100081:1-9.

Reply 7: We thank the learned reviewer for the suggestion. We have add the reference in the revised manuscript. (Page: 5, Line: 7)

Query 8: Line 92: The authors should also add this article to proof that chitosan was successfully used to target colitis: xxxoshi

Reply 8: We thank the learned reviewer for the suggestion. We have add the reference in the revised manuscript. (Page: 6, Line: 10)

Query 9: Line 123: Did the authors check the stability of their NPs in different pH (2.1 for stomach, 6.8 for small intestine and 7.4 for colon) similar to GIT conditions?

Reply 9: We thank the learned reviewer for his valuable suggestion.

As suggested by the learned reviewer we have check the aqueous stability of the nanoparticle at different pH by monitoring its absorbance at 280 nm (Supplementary Figure S3a).

The size and the zeta potential under different pH values has also been investigated. The data are included for the learned reviewer (Supplementary Figure S3b). The size and the zeta potential under different pH values has also been investigated. The data are included for the learned reviewer.

It has to be noted that the structural stability of the NPs in different pH remains same. Our TEM study in different pH confirms that there is no significant effect of pH in shape and size of the core nanoparticle (Supplementary Figure S3c-3f). The aqueous stability of the NP in different pH have also been studied. From the absorbance and zeta potential studies we can conclude that the synthesized NP is precipitating out from the solution at higher pH as chitosan molecules tends to aggregate at higher pH. So, this characteristic of our NP is very much useful in intestine targeted delivery which was the aim of this study. The issue of stability in different pH has been included in the revised manuscript. (Page: 8, Line Number: 3-14)

Query 10: Line 146: The author should write the full name of DCFH.

Reply 10: We thank the learned reviewer for the suggestion. We have addressed the issue in the revised manuscript.

सत्येन्द्र नाथ बसु राष्ट्रीय मौलिक बिज्ञान केन्द्र
**SATYENDRA NATH BOSE NATIONAL
CENTRE FOR BASIC SCIENCES**

Query 11: The authors should add the reference that they follow to prepare their nanoparticle.

Reply 11: We thank the learned reviewer for the suggestion. We have addressed the issue in the revised manuscript. (Page: 24, Line Number: 2-3)

Query 12: Lines 446, 437, 493 and 499, 515: The authors should add the references.

Reply 12: We thank the learned reviewer for the suggestion. We have add the references in the revised manuscript.

Reviewer: 3

Recommendation: Publish after minor revisions noted.

Comments:

This is a routine study. Physiological observations including fecal blood and defecation frequency, as well as elevated inflammatory molecular markers, established inflammatory UC in a mouse model after dextran sulfate (DSS) induction. The application of this nanomaterial greatly reduces ROS in the foci, and reduces the mortality of the disease, as well as the inflammation of the disease, the manuscript reflects many works, but the significance is not significant.

Reply: We would like to thank the learned reviewer for carefully reading the manuscript and appreciating our work. We have addressed all the queries point by point in the following section.

General comments:

Query 1: Material characterization and cytotoxicity hemolysis assays need to be supplemented.

Reply 1: We thank the learned reviewer for carefully reading your manuscript. Please note that we have provided the structural, chemical and spectroscopic characterisation of the material in the revised manuscript, as per the kind suggestion of the learned reviewer (Figure 1, Supplementary Figure S1, Supplementary Figure S2, Supplementary Figure S3, Supplementary Figure S4). The cytotoxicity of the material has also been assessed using HEK 293 cell line (Supplementary Figure S7, in the previous manuscript). We have already performed the sub-chronic toxicity study in preclinical mice following the well-recognized testing protocol of OECD (Supplementary Figure S8,

BLOCK-JD, SECTOR-III, SALT LAKE, KOLKATA-700098

Phones- 0091-(0)33-2335 5705-8, 2335 3057/61, 2335 0312/1313

E-mail: root@bose.res.in

Webpage: <http://www.bose.res.in>

Fax: 0091-(0)33-2335 3477

FUNDED BY THE DEPARTMENT OF SCIENCE & TECHNOLOGY, GOVT. OF INDIA

सत्येन्द्र नाथ बसु राष्ट्रीय मौलिक बिज्ञान केन्द्र
**SATYENDRA NATH BOSE NATIONAL
CENTRE FOR BASIC SCIENCES**

in the previous manuscript). However, according to your kind suggestion we have also performed the haemolysis assay and no toxic effect of the nanoparticles were observed in the RBCs (Page:18 Line:1-4, Figure 6b). As per the kind the suggestion of the learned reviewer, we have included a detailed discussion on the toxicity of the material in HEK 293 (Page: 16-17, Figure 6a)

Query 2: Lack of basic experiments and interventions to confirm the reliability of the experiments.

Reply 2: We appreciate the concern of the learned reviewer. However, please note that we have performed all the experiments to the best of our ability and have also repeated the experiments several times to gain a more robust statistical outcome regarding the results. We disagree with the learned reviewer regarding the reliability of the results as all the experiments were done and reviewed by experts on that field. However, still if the learned reviewer has queries regarding the authenticity and the reliability of the results, data is available from the corresponding author on reasonable request.

Query 3: Lack of confirmation of the effects of modified materials on other organs.

Reply 3: We would like to thanks the learned reviewer for his kind comments. We have already performed the sub-chronic toxicity study in preclinical mice following the well-recognized testing protocol of OECD (Supplementary Figure S8, in the previous manuscript). The histopathological findings shows no toxic effect to any organs (Supplementary Figure S8, in the previous manuscript). However, according to your kind suggestion, in the revised manuscript we have provided a main figure (Figure 6) consisting the data of the toxicity of the material for murine model. (Figure 6d, Page:18, Line:12- Page:19, Line:1-2)

Query 4: Application of nanomaterials the timing and trend of specific anti-inflammatory oxidative stress have not been identified.

Reply 4: We thank the learned reviewer for the thorough reading of your manuscript. Please note that we have checked the cytokines markers and proteins like NF- κ B, Caspase I. These markers are known to have direct correlation with the oxidative stress. Also the mitochondrial damage is directly associated with the oxidative stress. So, by quantifying these markers and observing the reversal of mitochondrial damage we have successfully established the modulation of oxidative stress by our nanoparticle in the physiological milieu.

Query 5: It is recommended to add some western blotting and PCR to enhance the reliability of the experiment.

Reply 5: We appreciate the concern of the learned reviewer. We disagree with the learned reviewer that western blotting and PCR doesn't increase the reliability of an

सत्येन्द्र नाथ बसु राष्ट्रीय मौलिक बिज्ञान केन्द्र
**SATYENDRA NATH BOSE NATIONAL
CENTRE FOR BASIC SCIENCES**

experiment. We have repeated each set of experiments to reach a statistically significant outcome. Please note that, we have performed Immunohistochemistry (IHC) study of colonic tissues to evaluate the expression of NF- κ B, Caspase-1 and CD68. We are of the strong opinion that IHC studies are adequate to prove the expression level of these proteins. Hence, western blotting was not performed. In addition, PCR technique is redundant in this current study to support the claims that we propose. Further, the quantification of specific ILs like IL-12 and IL-1 β were done using standard absorbance ELISA kit.

Other comments:

1. The quality of the numbers is hard to read, there are no numbers.
2. There are many grammatical errors in the manuscript. English should be edited.
3. It is recommended to change to other magazines.

Reply: We would like to thank the learned reviewer for carefully reading the manuscript. We have addressed all the queries in the revised manuscript to the best of our ability.

सत्येन्द्र नाथ बसु राष्ट्रीय मौलिक बिज्ञान केन्द्र
**SATYENDRA NATH BOSE NATIONAL
CENTRE FOR BASIC SCIENCES**

Title Page:

Chitosan Functionalized Mn₃O₄ Nanomedicine Counteracts Ulcerative Colitis through Modulation of Exposome

Susmita Mondal¹, Monojit Das^{2,3}, Ria Ghosh¹, Manali Singh⁴, Aniruddha Adhikari¹, Soumendhra Darbar⁵, Anjan Kumar Das⁶, Siddhartha Sankar Bhattacharya², Debasish Pal², Debasish Bhattacharyya⁷, Ahmed S. A. Ahmed⁸, Asim Kumar Mallick⁹, Munirah M. Al-Rooqi¹⁰, Ziad Moussa¹¹, Saleh A. Ahmed^{10,12*} Samir Kumar Pal^{1,2*}

¹Department of Chemical, Biological and Macromolecular Sciences,
S. N. Bose National Centre for Basic Sciences,
Block JD, Sector 3, Salt Lake, Kolkata-700106, India

²Department of Zoology, Uluberia College,
University of Calcutta,
Uluberia, Howrah-711315, India

³Department of Zoology,
Vidyasagar University,
Rangamati, Midnapore-721102, India

⁴Department of Biotechnology
Thapar Institute of Engineering and Technology
Bhadson Road, Patiala, Punjab147004, India

⁵Research & Development Division,
Dey's Medical Stores (Mfg.) Ltd,
62, Bondel Road, Ballygunge, Kolkata 700019, India

⁶Dept. Of Pathology,
Cooch Behar Government Medical College & Hospital
Vivekananda Rd, Khagrabari, Cooch Behar, West Bengal 736101

⁷Department of Gynecology & Obstetrics,
Nil Ratan Sircar Medical College & Hospital,
138, AJC Bose Road, Sealdah, Raja Bazar, Kolkata, 700014 India

⁸Faculty of Medicine, Assiut University,
71516 Assiut, Egypt

⁹Department of Pediatric Medicine,
Nil Ratan Sircar Medical College and Hospital,
38, Acharya Jagadish Chandra Bose Rd, Sealdah, Raja Bazar,
Kolkata, West Bengal 700014

BLOCK-JD, SECTOR-III, SALT LAKE, KOLKATA-700098
Phones- 0091-(0)33-2335 5705-8, 2335 3057/61, 2335 0312/1313
E-mail: root@bose.res.in

Webpage: <http://www.bose.res.in>

Fax: 0091-(0)33-2335 3477

FUNDED BY THE DEPARTMENT OF SCIENCE & TECHNOLOGY, GOVT. OF INDIA

सत्येन्द्र नाथ बसु राष्ट्रीय मौलिक बिज्ञान केन्द्र
**SATYENDRA NATH BOSE NATIONAL
CENTRE FOR BASIC SCIENCES**

¹⁰Department of Chemistry,
Faculty of Applied Science, Umm Al-Qura University,
21955 Makkah, Saudi Arabia

¹¹Department of Chemistry, College of Science, United Arab Emirates University, P.O. Box
15551, Al Ain, United Arab Emirates

¹²Department of Chemistry,
Faculty of Science, Assiut University,
71516 Assiut, Egypt

***Corresponding Authors**

E-mail Addresses: skpal@bose.res.in; saahmed@uqu.edu.sa
Telephones: (+91)33 2335 5706-08; (+966)530435760
Fax: (+91)33 2335 3477

Acknowledgements: The authors would like to acknowledge the Deanship of Scientific Research at Umm Al-Qura University, for supporting this work by Grant code: 22UQU4320545DSR08. Dr.Ziad Moussa is grateful to the United Arab Emirates University (UAEU) and to the Research Office for supporting the research developed in his laboratory and reported herein (SUREPLUS Grant code G00003918).

BLOCK-JD, SECTOR-III, SALT LAKE, KOLKATA-700098
Phones- 0091-(0)33-2335 5705-8, 2335 3057/61, 2335 0312/1313
E-mail: root@bose.res.in
Webpage: <http://www.bose.res.in>
Fax: 0091-(0)33-2335 3477

FUNDED BY THE DEPARTMENT OF SCIENCE & TECHNOLOGY, GOVT. OF INDIA

सत्येन्द्र नाथ बसु राष्ट्रीय मौलिक बिज्ञान केन्द्र SATYENDRA NATH BOSE NATIONAL CENTRE FOR BASIC SCIENCES

Author contributions

S.M., S.A.A. and S.K.P. designed the experiments. S.M., A.A., R.G. and M.S. did synthesis, characterization, and in vitro studies. S.M., M.D. and S.S.B performed cellular studies. S.M., A.A., S.D., and M.D. conducted animal experiments. A.K.D. performed the histological studies and analysis. S.M., A.A., D.P., D.B., A.S.A.M., A.K.M., M.M.A., Z.M., S.A.A., and S.K.P. discussed and analysed the results. S.M. wrote the manuscript and all authors contributed towards the writing of the final version.

Data availability:

The authors confirm that the all data are available in the main text and supplemental information. Any other related data that support the findings of this study are available from the corresponding author, S.K.P (skpal@bose.res.in), upon reasonable request.

BLOCK-JD, SECTOR-III, SALT LAKE, KOLKATA-700098
Phones- 0091-(0)33-2335 5705-8, 2335 3057/61, 2335 0312/1313
E-mail: root@bose.res.in
Webpage: <http://www.bose.res.in>
Fax: 0091-(0)33-2335 3477

FUNDED BY THE DEPARTMENT OF SCIENCE & TECHNOLOGY, GOVT. OF INDIA

REVIEWERS' COMMENTS:

Reviewer #1 (Remarks to the Author):

Authors have addressed issues raised by reviewer.
Current revised manuscript can be considered for publication

Reviewer #3 (Remarks to the Author):

This is a very novel article. The author used ChitosanFunctionalized Mn₃O₄Nanoparticle to treat DSS induced UC mice and improve their intestinal inflammatory symptoms. The author elaborates on three aspects: buffer solution, animal model, and cell. The inhibitory effects of NP on ROS and downregulation of inflammatory mediators were demonstrated. In addition, the author also elucidated the mechanism by which NP inhibits inflammation by protecting mitochondria. The entire article is well organized and detailed. However, in terms of NP safety, it is recommended to add heart and lung sections as well as some biochemical indicators. This will be more comprehensive and objective.

Reply to the Reviewers

Authors would like to thank the learned reviewers for their very critical comments which defiantly help to enhance the quality of presentation in revised manuscript. In the following section we have addressed the issues raised by the learned reviewers point by point.

Reviewer #1

Comment 1:

Authors have addressed issues raised by reviewer.

Current revised manuscript can be considered for publication

Reply 1: We would like to thank the learned reviewer for appreciating our work.

Reviewer #3

Comment 1:

This is a very novel article. The author used Chitosan Functionalized Mn_3O_4 Nanoparticle to treat DSS induced UC mice and improve their intestinal inflammatory symptoms. The author elaborates on three aspects: buffer solution, animal model, and cell. The inhibitory effects of NP on ROS and downregulation of inflammatory mediators were demonstrated. In addition, the author also elucidated the mechanism by which NP inhibits inflammation by protecting mitochondria. The entire article is well organized and detailed. However, in terms of NP safety, it is recommended to add heart and lung sections as well as some biochemical indicators. This will be more comprehensive and objective.

Reply 1: We would like to thank the learned reviewer for appreciating our work. We have added the histopathological images of heart and lung treated with our nanodrug. No toxicity was found in the section of heart and lung. This has been discussed in the revised manuscript (Page 18-19, Line Number: 385-388, Figure 6d).

As biochemical indicator, we have also added the differential count of WBC in the supplementary table 1. No toxicity was found in the differential count of WBC. This has been discussed in the revised manuscript (Page:19, Line Number: 391-399)